# Unwinding of a DNA replication fork by a hexameric viral helicase

Abid Javed [1,3], Balazs Major[2,3], Jonathan A. Stead [2,3], Cyril M. Sanders [2✉] & Elena V. Orlova [1✉]

Hexameric helicases are motor proteins that unwind double-stranded DNA (dsDNA) during DNA replication but how they are optimised for strand separation is unclear. Here we present the cryo-EM structure of the full-length E1 helicase from papillomavirus, revealing all arms of a bound DNA replication fork and their interactions with the helicase. The replication fork junction is located at the entrance to the helicase collar ring, that sits above the AAA + motor assembly. dsDNA is escorted to and the 5′ single-stranded DNA (ssDNA) away from the unwinding point by the E1 dsDNA origin binding domains. The 3′ ssDNA interacts with six spirally-arranged β-hairpins and their cyclical top-to-bottom movement pulls the ssDNA through the helicase. Pulling of the RF against the collar ring separates the base-pairs, while modelling of the conformational cycle suggest an accompanying movement of the collar ring has an auxiliary role, helping to make efficient use of ATP in duplex unwinding.

[1] Department of Biological Sciences, Birkbeck College, Institute of Structural and Molecular Biology, Malet Street, London WC1E 7HX, UK. [2] Academic Unit of Molecular Oncology, Department of Oncology and Metabolism, University of Sheffield, Medical School, Beech Hill Rd., Sheffield S10 2RX, UK. [3]These authors contributed equally: Abid Javed, Balazs Major, Jonathan A. Stead. ✉email: c.m.sanders@sheffield.ac.uk; e.orlova@mail.cryst.bbk.ac.uk

DNA replication is an essential process in all living organisms. It starts at specific sites known as origins of replication (*ori*), where helicase enzymes begin the unwinding of double-stranded DNA (dsDNA), generating replication forks (RFs) that grow bi-directionally from *ori*[1]. Helicases use the energy of nucleoside triphosphate (NTP) hydrolysis to translocate and unwind DNA, providing the single-stranded template (ssDNA) for accurate copying by DNA polymerase. Four of the six known helicase superfamilies (SF3–6) are enzymes assembled from six subunits arranged as hexameric rings, while those of SF1 and 2 are monomeric but sometimes function as dimers[2,3]. In cells, the principal replicative helicase is a hexamer, but despite their crucial function how these helicases unwind dsDNA remains uncertain.

The prokaryotic SF4 and SF5 helicases have a RecA fold NTP*ase* domain and translocate in the 5′–3′ direction on single-stranded nucleic acids[4,5]. In contrast to these, the NTP*ase* motor domain of the SF6 and viral SF3 helicases have an AAA+ (ATP*ase* Associated with various Activities) fold[6,7] and translocate in the 3′–5′ direction[4]. Most hexameric helicases including papillomavirus E1, bacteriophage T7 gp4, *E. coli* DnaB and archaeal MCM are homo-oligomers, while the eukaryotic Mcm2-7 AAA+ hexamer, that forms the core of the CMG (Cdc45-MCM-GINS) helicase complex, is composed of six related but non-identical subunits[4,7,8]. These helicases are assumed to operate by a strand (or steric) exclusion mechanism[5]; translocating on the active nucleic acid strand the helicase moves towards the RF junction (RFJ) and unpairs the DNA bases, while the passive strand is excluded from the complex. Whether a hexameric helicase acts simply as a non-specific wedge or employs a specific separation pin[9,10] or other functional domains[11] to optimise base separation, as in SF1 and SF2 helicase, is unclear.

Crystal structures of homo-hexameric helicases with short single-stranded nucleic acid (NA) segments bound in the NTP*ase* motor domain have been obtained for DnaB[12], the RNA helicase Rho[13], the helicase domain of E1[14], and archaeal MCM[15]. These structures show the nucleotides of the NA chain interacting with a "spiral staircase" of binding loops in the protein complex. Accordingly, mechanisms for NA translocation based on the sequential hydrolysis of ATP and a cyclical height-adjusted movement of the NA-binding loops have been suggested. Recent cryo-electron microscopy (cryo-EM) structures of yeast[16–18], *Drosophila*[19,20] and human[21] MCM complexes bound to fork-like DNA substrates are also consistent with this general translocation mechanism. These structures, however, did not reveal the DNA unwinding point and interactions of the helicases with all arms of the RF in detail, so how DNA strand separation is achieved remained unclear.

The papillomaviruses (PVs) are a large group of human and animal pathogens[22] and the PV E1 protein is a model AAA + SF3 hexameric helicases[6,23]. The N-terminal half of PV E1 has a regulatory module and a sequence-specific dsDNA-binding domain (OBD) for PV *ori* DNA recognition (Fig. 1a, bovine papillomavirus (BPV) E1). A hexamer of the C-terminal helicase domain (E1HD) can function alone to unwind dsDNA in vitro[24]. The E1HD subunit can be sub-divided into the collar domain and the NTP*ase* motor domain that form rings in the E1HD structure[14,25]. Interestingly, E1, like Mcm2-7, can encircle both dsDNA and ssDNA[26,27]. Also, the N-terminal domains of these proteins are located in front of the helicase motor, in the vicinity of the RFJ[16–21,28]. In a low-resolution EM structure of the full-length E1 helicase bound to a synthetic DNA RF the unwinding point was mapped at the entrance of the helicase collar domain[29], consistent with the steric exclusion model. However, the structure also showed extensive interactions of the N-terminal domains of the protein with the DNA ahead of the replication fork.

Here we present a near-atomic resolution (3.9 Å) cryo-EM structure of E1 revealing clearly all arms of the RF and the dsDNA unwinding point for an AAA+ hexameric helicase. Previous structures of the eukaryotic CMG helicase complex bound to fork substrates have been determined at resolutions close to 4 Å[16–21], providing important insights into the mechanism of CMG catalysed DNA unwinding. However, no structure has revealed the replication fork in its entirety so mechanistic understanding is limited. In the E1 helicase, dsDNA is separated at the entrance to the collar domain ring (Fig. 1a, b), as the AAA+ motor pulls one ssDNA strand through the complex and the RFJ against the collar ring. Two of the six E1 OBDs are observed at fixed positions, where one escorts the dsDNA to the RF unwinding point and the other the unwound 5′ssDNA away from the complex. We have also been able to trace the C-terminal acidic tails for all six E1 subunits, explaining their functional role in processive unwinding. The structure also shows deviations in the positions of the collar domains and flexibility of the AAA+ domains of the helicase induced by the presence of the RF, providing evidence that the collar is actively employed in strand separation.

## Results

**Cryo-EM structure of the E1RF complex.** The crystal structures of the E1HD with[14] and without[25] ADP and ssDNA bound (PDB 2GXA, 2V9P, respectively) are both asymmetric hexameric assemblies and show similar nucleotide and ssDNA-binding site architecture, despite the presence or absence of ligands. Accordingly, we reasoned that a stalled E1RF assembly could be generated in the absence of nucleotides and without using artificial DNA roadblocks[16] to impede translocation. A hexameric E1-replication fork (E1RF) complex stalled at a RFJ was assembled and purified as previously described (Fig. 1c)[29]. The DNA replication fork (RF) substrate consisted of 30 base pairs of dsDNA, a 3′ T20-active ssDNA strand (the strand upon which the helicase translocates, or the leading DNA replication strand) and a 13 base 5′ passive (lagging replication) strand (Fig. 1b, see the "Methods" section). The fork substrate used is actively unwound by the helicase, while substrates lacking a 5′ passive strand are less efficient and substrates without an active 3′ strand are not unwound significantly[29] (Fig. 1d).

Cryo-EM data of E1RF complexes were collected on a Titan Krios microscope operating at 300 keV using a Gatan K3 camera and were processed using RELION 3.0[30] and cryoSPARC v2.1[31] (Supplementary Table 1 and Supplementary Figs. 1 and 2, see the "Methods" section). The 2D class averages of cryo-EM images of the E1RF showed a distinct, two-tiered structure confirmed to be the collar and AAA+ domains of the E1HD module (Fig. 2a, Supplementary Fig. 1). There is a ~20 Å wide rod of density extending from the E1HD, slightly tilted with respect to its central axis (17°), with a bulk of density on its outer side (Fig. 2). There is also density in the central channel of the E1HD, indicating ssDNA binding and therefore a stable complex with the DNA fork has been formed. Diffuse density visible above the E1HD region in class averages suggests a flexible arrangement of some OBDs and N-terminal sub-domains (Supplementary Figs. 1a and 3).

A cryo-EM E1RF map was obtained at a resolution of 3.9 Å (Fig. 2, Supplementary Figs. 1 and 2). The structure demonstrates unambiguously the positions of the dsDNA, the 3′ssDNA active strand within the entire central channel of the E1HD and the 5′ ssDNA passive strand, positioned on the top of the collar ring (Fig. 2). The atomic model of dsDNA could be superposed with the rod of density protruding from the centre of the hexamer. The dsDNA is contiguous with the 3′ ssDNA strand located in the central E1HD ssDNA binding tunnel and the 5′ passive ssDNA

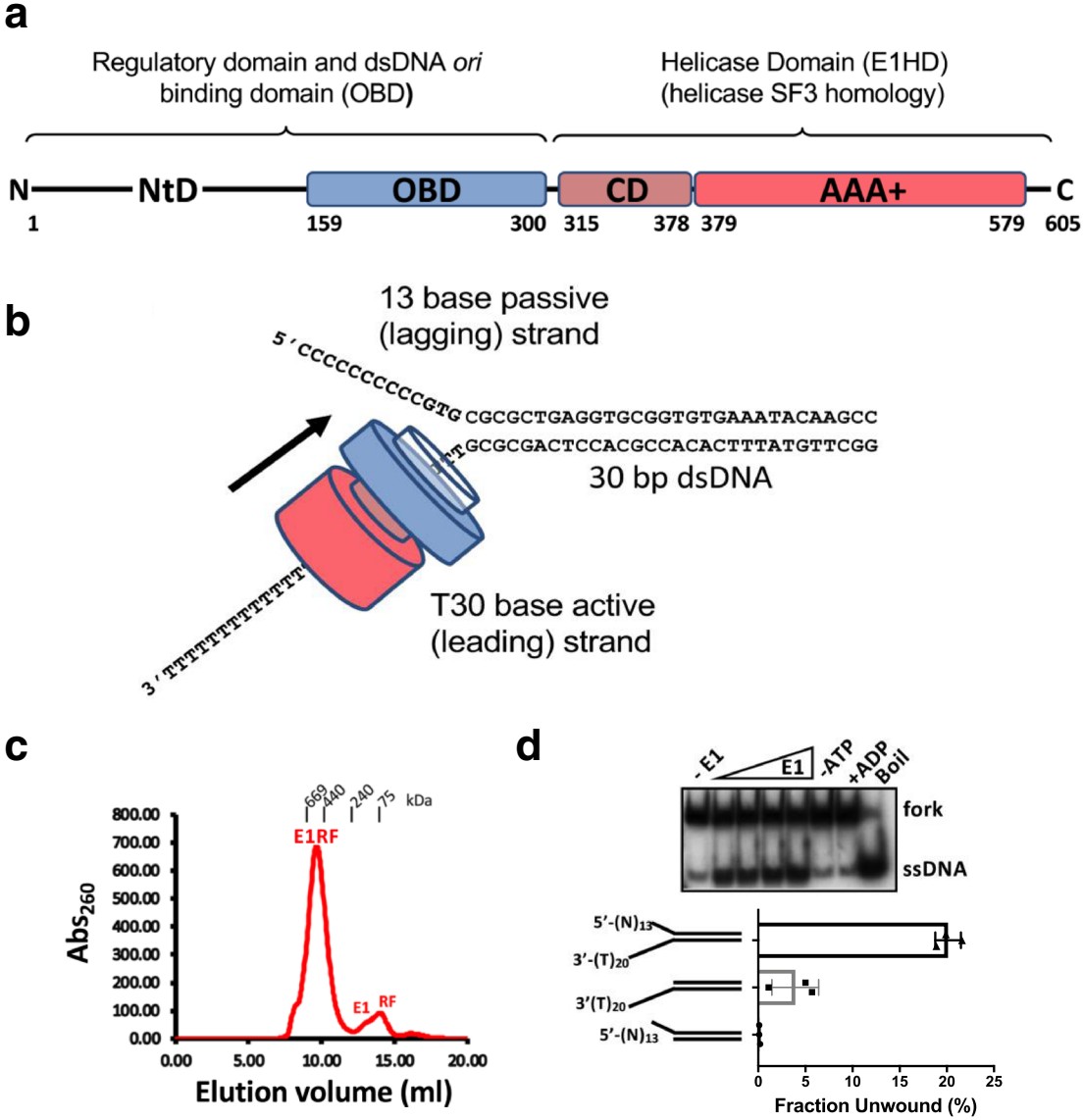

**Fig. 1 The E1RF helicase complex. a** Domain organisation of Bovine Papillomavirus (BPV) E1. **b** A schematic representation of the E1RF complex on the fork sequence used. **c** A homogenous fraction of the E1RF complex purified by gel filtration was used for cryo-EM. The calibration markers are thyroglobulin, 669 kDa, ferritin, 440 kDa, catalase 240 kDa and conalbumin, 75 kDa. **d** The unwinding activity of the E1 protein (50, 100, 200 and 400 nM E1) on the replication fork substrate used to generate E1RF and comparison with substrates without 5′ or 3′ ssDNA tails (100 nM E1, $n = 3$ independent experiments; shown mean values with error bars ± SD).

strand (Fig. 2). The dsDNA separation point, the RFJ, is located at the entrance to the collar tunnel. Based on the positions of the ssDNA-binding β-hairpins we aligned the E1RF structure with respect to the crystal structures[14,25], as described below. For direct comparison, we labelled the six protein subunits of the E1RF assembly A–F, corresponding to the subunit designation in the E1HD crystal structures.

Notably, the cryo-EM structure has two well-defined bulks of densities in addition to the E1HD and DNA. One is located ~25 Å above the E1HD collar ring and is attached to the dsDNA, while the second, on the opposite side, is at a lower position bound to the 5′ssDNA (Fig. 2a). Density corresponding to the E1 NtDs (Fig. 1a) was not resolved in the EM map suggesting that these parts of the complex are disordered. Significantly, the densities for the C-terminal acidic tail of E1 (C-tT, residues 579–605, Fig. 1a) were defined for all six subunits up to residue 598, but modelled as poly-alanine. This domain is required for hexamer stability and

processive DNA unwinding[32]. The residues of the C-tT, while present in the protein construct used to obtain the nucleotide and DNA free E1HD structure (but absent in the ligand-bound form[14]), were not visible in the X-ray map[25].

**Positions of the OBDs.** The two additional bulks of high density correspond very well to the size of the OBD[33] (PDB 1KSX and 1KSY). The orientations of these OBDs, from the B and E sub-units (Fig. 2b), were defined by the links between their N-termini and the corresponding C-termini of the E1HD collar domain. OBDs B and E were better defined due to their interactions with DNA (Supplementary Fig. 3). The OBD of subunit B interacts with the dsDNA while the OBD from subunit E interacts with the 5′ passive ssDNA strand and is located close to the side of the E1HD collar ring (Fig. 2). The other OBDs and their associated NtDs of the other subunits were defined less well, to varying degrees, but they surround the dsDNA (Supplementary Fig. 3).

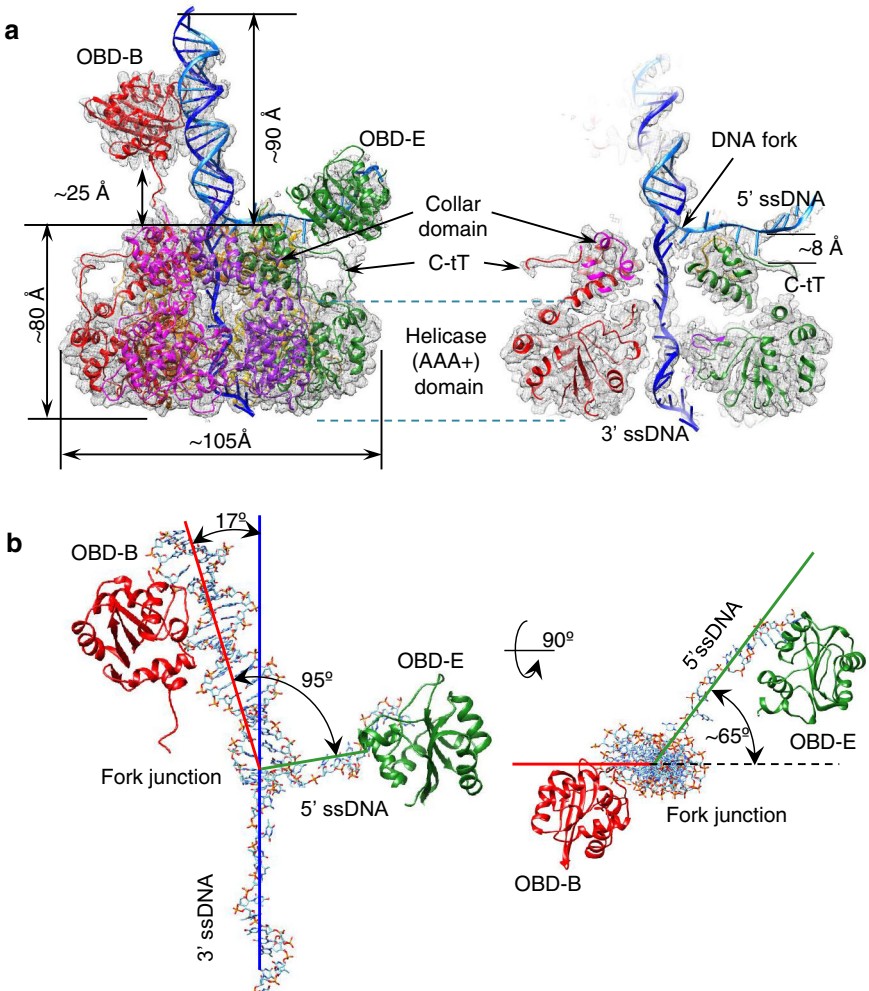

**Fig. 2 The E1RF structure. a** Side view (left) and vertical central slice (right) of the E1RF EM map (in grey), with the fitted atomic models of the RF DNA (blue) and E1HD and OBD domains. Subunits A–F of the E1RF are shown in pink, red, orange, yellow, green, and purple correspondingly (and in all subsequent figures). **b** Fitted atomic model of the unwinding replication fork and OBD B and OBD E interacting with the dsDNA and 5′ ssDNA strands, respectively, viewed from the side (left) and top (right). The angles between the dsDNA and the 5′ ssDNA are indicated.

Analysis of OBDs A, C, D and F revealed significant flexibility in their positions, with the locations of A and C least defined.

**Interaction of E1 OBD B with dsDNA.** X-ray structures of the E1 OBD showed that it has two DNA-binding segments, a DNA-binding loop (DBL, Arg180–Asn189) and a DNA-binding helix (DBH, Arg243–Leu254), that recognise dsDNA-binding sites at the PV *ori*, related to the E1 binding site (E1BS) consensus sequence 5′-ATTGTT[33,34]. The DBL makes the major contribution to dsDNA binding through generic hydrophilic and van der Waals contacts, consistent with the relatively low binding specificity and affinity of the interaction. In the E1RF structure, an automated docking (iMODFIT[35], see the "Methods" section) of the E1 OBD atomic model (1KSY) into OBD B of the EM map resulted in an RMSD of 3.6 Å (Fig. 3a, b). While the dsDNA sequence of the fork does not contain an E1BS-like sequence, the cryo-EM structure indicates that OBD B interacts with the dsDNA via an interaction between the DBL and the major groove at the sequence TGTGA in the passive DNA strand, 16-20 nucleotides from the RFJ (Fig. 3c). Together, the well-defined OBD B-dsDNA interaction and contacts with the other surrounding, but more loosely positioned, OBDs (Supplementary Fig. 3) are consistent with previous biochemical "footprinting" experiments[29]. This analysis demonstrated protection of the

dsDNA from hydroxyl radical nucleolytic attack, most likely by direct contact with the OBDs and N-terminal segments of E1.

Lysines 183 and 186 in the OBD DBL are conserved in papillomavirus sequences (Supplementary Fig. 4) and have been demonstrated to be critical for dsDNA binding[36]. To test if dsDNA interactions with the OBD influence helicase activity we generated an E1 protein with alanine at positions 183 and 186 (K183A/K186A). In helicase assays (Fig. 4a), a nearly two-fold reduction in DNA unwinding was observed for the altered protein (Fig. 4b, Supplementary Fig. 5a), demonstrating that the OBD domain has an auxiliary role in DNA unwinding.

**The collar domain ring.** In the E1RF structure, the subunits of the collar ring are arranged with nearly six-fold rotational symmetry and three nucleotides of ssDNA appear stretched through its channel (Fig. 2a). The collar ring of E1RF is rigid and superposition with the crystal structure of E1HD/ssDNA/ADP (PDB 2GXA)[14] indicates a low overall structural deviation (RMSD 0.65). In the E1RF structure, the conserved positively charged residues Lys356 and Lys359 project their side chains into the E1HD channel but their ε-amino groups are at least 4 Å from the 5′ ssDNA phosphate backbone (Fig. 5a, b). As such, strong electrostatic interactions between protein and ssDNA are unlikely, as supported by observations that substitution of these

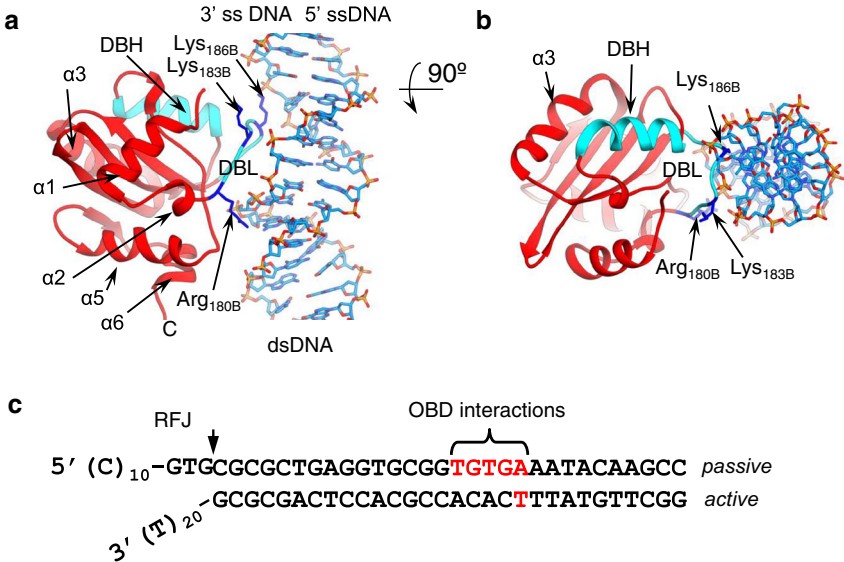

**Fig. 3 Interaction of OBD B with dsDNA. a** The fitted atomic model for OBD-B (in red, α-helices labelled) with the DBH (Thr239-Asn248) and the DBL (Arg180-Asn189) shown in cyan. The DBL interacts with the dsDNA. **b** The interaction of OBD B is viewed from the top showing Lys183 and Lys186 interacting with the dsDNA. **c** The interaction of OBD B with dsDNA projected on the fork sequence, interacting sequence highlighted turn in red.

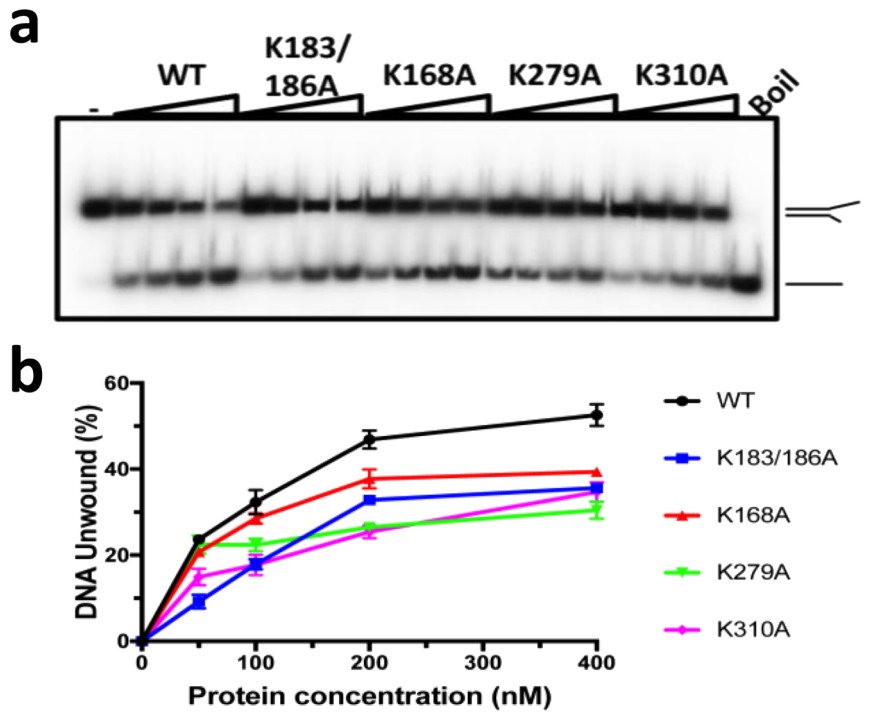

**Fig. 4 Unwinding activity of E1 mutants.** Mutations were generated to disrupt interactions with the dsDNA (double mutant K183A/K186A) and 5′ ssDNA (K168A and K279A in the OBD and K310A in the inter-domain linker) of E1. **a** Purified proteins (50, 100, 200 and 400 nM) were assayed using a fork-like substrate (0.1 nM) with the same sequence used to form E1RF. Boil is the thermally denatured substrate and '-' is the native substrate (no E1). **b** Graphical representation of the data ($n = 3$ independent experiments; shown mean values with error bars ± SD). Wild-type (Wt), black line; K183/K186A, blue; K168A, red; K279A, green; K310A, magenta.

residues has no significant effect on ssDNA binding and unwinding[37].

**Exit of the 5′ passive ssDNA strand.** In the E1RF cryo-EM structure, the 5′ ssDNA is diverted from the separation point at an angle of 95° relative to the dsDNA axis (Fig. 2b), passing in a groove between E1 subunits D and E (Fig. 5a). The elements of the collar domains that cradle the RFJ are the loop residues

351ThrAsnSer353 (TNS loop) from an α3–α4-hairpin turn in E1 collar domain subunit D. DNA footprinting experiments show that the DNA at the RFJ is protected from nucleolytic attack, implying close protein–DNA contacts[29]. However, there is no evidence that these hairpins are involved directly in dsDNA unwinding as this takes place above the hairpins (~4 Å) and the distances between the RFJ and the α–α-hairpin turn is therefore rather large.

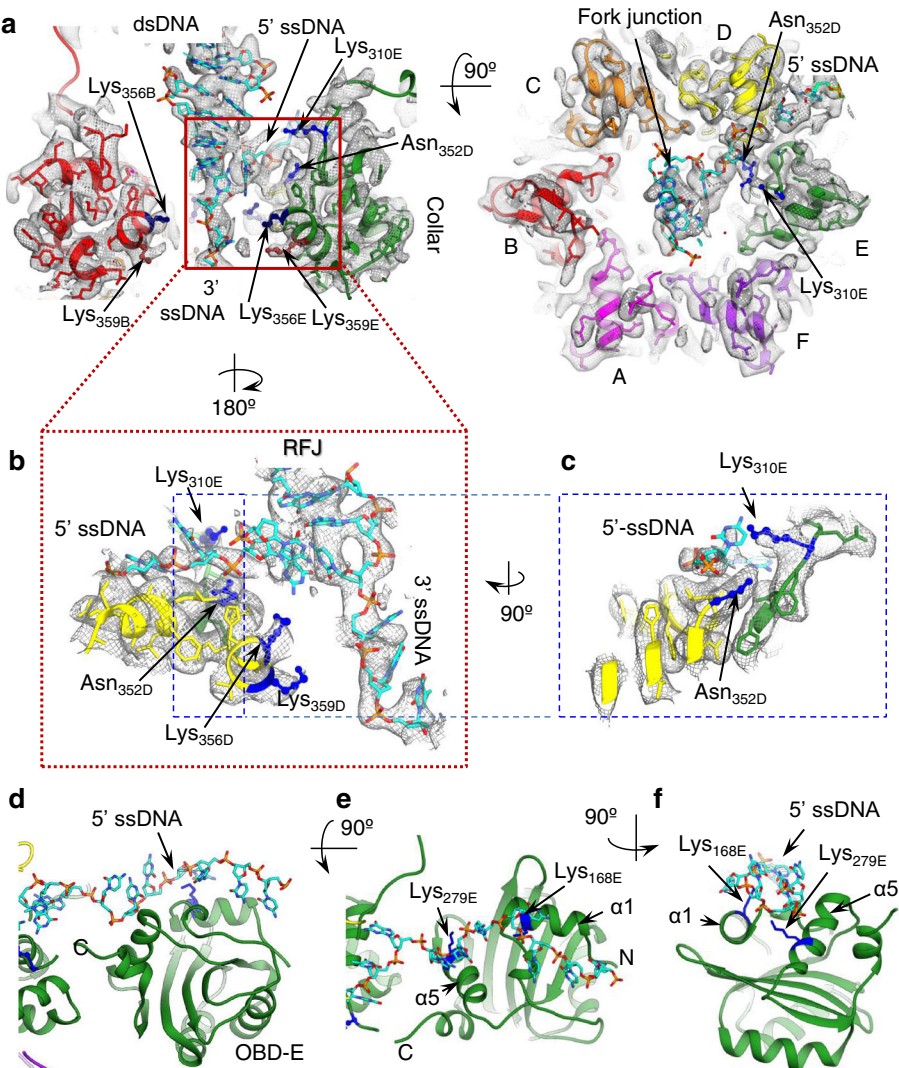

**Fig. 5 The exit path of the 5′ passive ssDNA strand. a** Fitted atomic model of the DNA fork junction in the EM map (grey mesh). Subunit D Asn352 and subunit E Lys310 are indicated. Residues Lys356 and Lys359 are highlighted in dark blue and brown correspondingly. **b** and **c** Atomic models fitted into the EM density map to show the interactions of Asn352 and Lys310 with the ssDNA close to the replication fork junction (RFJ). Locations of residues Lys356 and Lys359 in subunit D are shown in blue. **d–f** The atomic model of the interaction of the 5′ ssDNA with the subunit E OBD, with Lys168 and Lys279 indicated.

Four distinct features mark the route taken by the 5′ ssDNA strand. First, the ssDNA is close to Lys310 in the inter-domain linker between the OBD and E1HD (distance ~2.3 Å, subunit E), which makes a contact with the ssDNA likely, thus fixing the path of the ssDNA (Fig. 5b, c, Supplementary Fig. 5b). While Lys310 is not conserved in the papillomavirus sequences (Supplementary Fig. 4) the K310A mutation shows a significant reduction (up to ~50%) in helicase activity, supporting its functional role in BPV E1 DNA unwinding (Fig. 4a). Second, the TNS loop of subunit D is interacting with and displacing the ssDNA upward (Fig. 5a–c). Alignment of nearly 100 papillomavirus sequences from the databases reveals an overwhelming preference for polar residues in this segment; only in a few cases is Asn352 substituted with serine, threonine and very rarely alanine. A range of amino acid substitutions tested at position 352 all showed reduced DNA unwinding activity (Supplementary Fig. 5c–e). Notably, glycine and lysine substitutions showed more than 50% and a nearly four-fold reduction of unwinding, respectively, indicating that relatively weak interactions with DNA by the polar Asn352 may be required. Together, therefore, Lys310 and Asn352 may be

optimal for guiding the exit path of the 5′ ssDNA. Third, we traced as poly-alanine the E1 C-terminal tails (C-tT). They start from the AAA+ domain and form loops at the subunit interfaces (Fig. 2a), ending within a cleft between collar domain subunits. This puts the acidic portion of the tail (amino acids 584–594) ~8 Å below the ssDNA path (Fig. 2a, left panel, and Supplementary Fig. 5b). The proximity of this segment to the ssDNA would induce repulsion between these electronegative elements, thus ensuring unimpeded passage of the ssDNA away from the helicase domain subassembly. Finally, the OBD from subunit E interacts with the 5′ passive ssDNA strand. The linker (residues 303–314) anchoring OBD E to the E1HD allowed us to define its approximate orientation, which was refined by an automated docking of the X-ray atomic model (1KSY) (Fig. 5d–f, see the "Methods" section). The fitting indicates that Lys168 of the N-terminal helix (α1) and Lys279 from helix α5 of the OBD are close to the ssDNA (3 and 4 Å, respectively), thus presenting a different binding surface to DNA compared to OBD B. Lys168 is conserved in all papillomavirus E1 sequences except for rare substitutions with arginine, while the majority of PV sequences

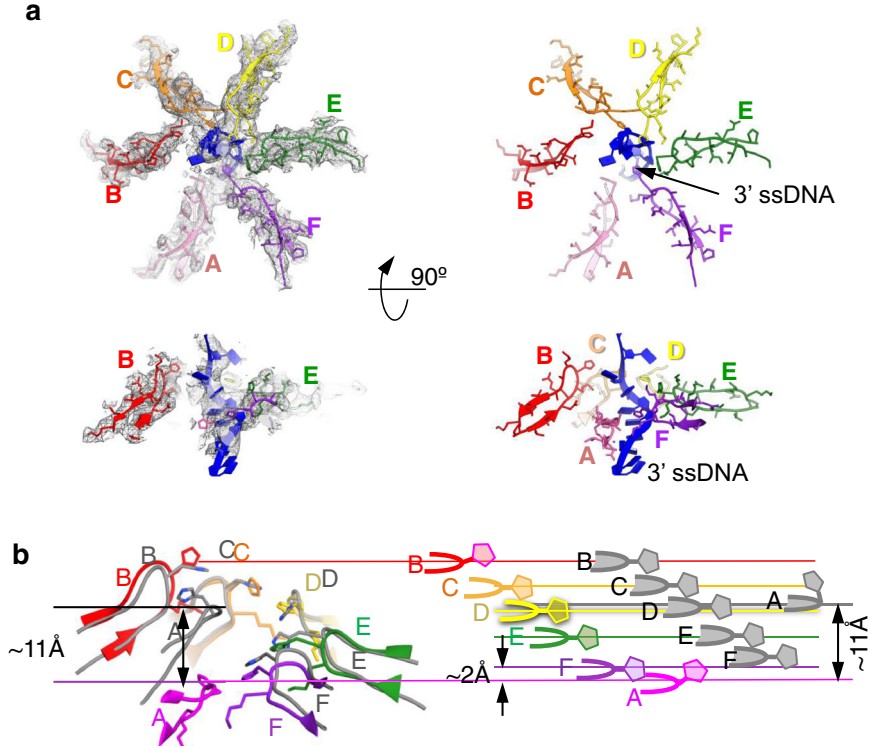

**Fig. 6 Interactions of the DNA-binding β-hairpins with the 3′-ssDNA. a** Top left, a cross-section of E1RF viewed from the top showing the fitted atomic model of the six DNA-binding β-hairpins in the EM map (grey mesh). The bottom left panel displays a side view of the same area revealing the staircase organisation of the β-hairpins. The right panels show the same slices of the atomic models with amino acid side chains to show interactions with the 3′ ssDNA. **b** Overlay of the DNA-binding β-hairpins from the fitted atomic model of the E1RF cryo-EM map (in colours corresponding to the E1 subunits) and the X-ray structure of E1HD/ssDNA/ADP[14] (in grey, PDB 2GXA). On the right, a cartoon illustrates the step-wise organisation of the individual DNA-binding β-hairpins in each structure. His507s are shown and their relative heights within the central channel of the AAA+ domain.

have Lys or Arg at the position corresponding to Lys279 in BPV E1 (Supplementary Fig. 4). In helicase assays, the variant E1 proteins K168A and K279A show ~15% and 30% reductions in DNA unwinding, respectively (Fig. 4a, Supplementary Fig. 5a). Accordingly, Lys168 and Lys279 of OBD E may play a role in guiding the emerging 5′ ssDNA strand away from the helicase motor.

**Interaction of the 3′ ssDNA with the E1HD.** In both E1HD crystal structures (PDB 2GXA with[14] and 2V9P without[25] cofactors bound) the ssDNA-binding segments of the subunits are positioned in a helical array along the axis of the ssDNA-binding tunnel. In E1HD/ssDNA/ADP (PDB 2GXA), interactions with ssDNA are mediated via the conserved DNA-binding β-hairpin (500–514 aa) residues Lys506 and His507[14,38]. The 2GXA structure has a large gap between subunits A and F and the β-hairpin of subunit A is positioned at the top of the staircase. In the E1RF complex, as in E1HD/ssDNA/ADP, six nucleotides of the 3′ ssDNA strand form a right-hand helix with the β-hairpins of the AAA+ domains, before it exits from the channel (Figs. 2a, 6a). In E1RF, the DNA-binding β-hairpins of subunits C–E are in contact with the DNA and align well with the 2GXA structure, while the β-hairpin of F also contacts DNA in E1RF it sits slightly below the corresponding β-hairpin in 2GXA. However, there is a difference in conformations to the corresponding β-hairpins in the A and B subunits. While in both structures the β-hairpin of subunit A does not make direct contact with the 3′ ssDNA, in E1RF the β-hairpin of A is positioned 11 Å below the corresponding β-hairpin in the crystal structure (Fig. 6b). The β-hairpin of subunit B does not contact the 3′ ssDNA either, since

its His507 is turned away from ssDNA compared to the crystal structure (Fig. 6a, b). Our observations, therefore, appear to be consistent with the coordinated escort mechanism for ssDNA translocation[14], although a different conformational state appears to be captured in the cryo-EM structure, where the subunit A β-hairpin has disengaged from ssDNA and has not yet migrated back to the top of the complex to re-engage with ssDNA.

**Biochemical analysis of E1RF–DNA interactions.** We analysed E1RF–DNA interactions using a footprinting assay, where close protein–DNA contacts are revealed by the protection of the DNA from hydroxyl radical (OH•) nucleolytic attack[29]. Hexameric helicase complexes were assembled with $^{32}$P end-labeled substrates, complete DNA binding was confirmed by gel-shift analysis, while the remainder of the reaction was exposed to the OH• (see the "Methods" section). The wild type E1RF complex was compared to assemblies with E1 K183A/K186A to probe OBD B interaction with the dsDNA and E1 K310A/K168A/N352G (targeting residues in the inter-domain linker, OBD E and collar domain, respectively; Figs. 4, 5 and Supplementary Fig. 5f) to probe the interaction with the 5′ ssDNA component of the RF substrate. Importantly, the homohexameric nature of E1RF does not allow distinct single-subunit interactions to be probed biochemically.

The E1RF OH• footprints (Supplementary Fig. 6), visualised and quantitated by phosphorimaging, show moderate and incomplete protection throughout the DNA, as would be expected for interactions that are extensive, but weak and transient. When the 3′ active strand of wild-type and E1 K183A/K186A RF complexes are compared (Supplementary

Fig. 6b, d), the ssDNA nucleotides close to the unwinding point show very similar levels of protection, while all other 3′ ssDNA nucleotides show increases in peak height (susceptibility to OH• cleavage) of up to ~12%. However, enhanced susceptibility to OH• cleavage is seen in the dsDNA of E1 K183A/K186A-RF, increasing with the distance from the RFJ (up to ~30% increase in peak height), implying weaker contacts between protein and dsDNA. On the 5′ passive DNA strand, the peak heights for the ssDNA cleavage products are nearly identical. However, again, peak heights increase by up to ~30% in the dsDNA region 10–25 nucleotides from the RFJ for E1 K183A/K186A compared to wild-type RF complexes. The diminished protection of the dsDNA in E1 K183A/K186A-RF appears most pronounced in the region ~15–20 bases from the RFJ, observed to interact with OBD B in E1RF (Fig. 3). These observations support the structural data showing that OBDs A–D and F surround the dsDNA (Supplementary Fig. 3), with B forming more stable contacts with dsDNA (Fig. 3).

For E1 K310A/K168A/N352G-RF the OH• cleavage pattern is different to K183A/K186A and wild-type RF complexes. For the 3′ active DNA strand (Supplementary Fig. 6b, e), the peak heights for cleavage products in the dsDNA nucleotides 6–25 positions from the RFJ are near-equivalent between variant and wild-type E1RF. However, peak heights decrease, implying tighter contacts with DNA, in the five dsDNA nucleotides close to the RFJ and up to nine 3′ssDNA nucleotides closest to the RFJ in E1 K310A/K168A/N352G-RF. It could be suggested that the positioning of the RFJ has been perturbed in this variant. In the 5′ passive DNA strand, cleavage of all dsDNA nucleotides and the three 5′ ssDNA nucleotides close to the RFJ are near equivalent (Supplementary Fig. 6e) for mutant compared to wild-type complexes. However, the 5′ ssDNA nucleotides at positions 4–7 from the RFJ show a subtle increase in protection for E1 K310A/K168A/N352G-RF. These observations suggest that residues Lys310, Lys168 and Asn352 minimise stabilising contacts with protein and the 5′ ssDNA, but are necessary for chaperoning the 5′ssDNA away from the protein complex during unwinding. Furthermore, the path taken by the 5′ ssDNA across the collar is predominantly neutral in character (Supplementary Fig. 5b), suggesting that there are in general no strong 5′ ssDNA interactions with the collar ring.

**Conformational changes in E1HD**. Although the collar ring is rigid, the alignment between the EM structure and the E1HD/ssDNA/ADP (PDB 2GXA) crystal structure shows changes in the positions of the collar domain subunits, where D and E are moved by ~3 Å up towards the 5′ ssDNA (Fig. 7a, and Supplementary Fig. 7). As such, the collar ring is tilted by 3° as a rigid body relative to its position observed in both X-ray structures[14,25]. In the AAA+ domains, however, translational shifts of some segments are significant, particularly in the A, B and F subunits, with Cα deviations between E1RF and the crystal structure 2GXA of up to 8 Å. These differences, observed mainly as shifts in the β-layers and α-helices at the periphery of the complex (Fig. 7, Supplementary Fig. 8), are likely to be observable due to the bound RF and natural non-crystallographic environment of E1RF in cryo-EM. In the coordinated escort model[14], the six AAA+ domains and their associated ssDNA-binding segments follow a conformational wave around the complex during ssDNA translocation (Fig. 7b). Interestingly, the α-5 helix of the E1HD appears to act as a main 'hinge' between subunit collar and AAA+ domains. In the E1RF structure, the α-5 'hinge' appears to move up to 6 Å in a wave-like trajectory around the hexamer subunits A–F (Fig. 7c). As such, these observations imply that the α-5 hinge motion would coordinate the movement of both the

collar and AAA+ domains during translocation on ssDNA (Fig. 7d and Supplementary Movie 1), with the tilt and elevation of the collar ring following the wave-like motion around the subunits to push up against the RFJ. Accordingly, ssDNA translocation and base-pair separation by the E1 helicase are coupled. We propose that as each of the six subunits of the E1 hexamer completes a cycle of ATP hydrolysis, pulling six ssDNA nucleotides through the AAA+ motor, there is an integral power stroke pushing against the RFJ, equivalent to the displacement of one base pair (~3 Å). We propose that the E1 helicase collar can be viewed as an active mechanical separation wedge, governed by nucleotide binding and hydrolysis events, rather than a simple obstacle for strand displacement.

**Discussion**

The crystal structure of E1HD bound to ssDNA and ADP[14] first provided a hypothesis for how hexameric helicases can translocate on single-stranded nucleic acids[12–21]. The E1RF cryo-EM structure now shows how E1 separates DNA base pairs and is optimised as a DNA unwinding machine (Fig. 8, Supplementary Movie 2).

Remarkably, the conformation of the RF DNA in E1RF is very similar to that observed in other DNA unwinding machines[39,40], suggesting its organisation is optimal for base separation. The conformation of the DNA fork in E1RF is maintained in several ways. First, the E1 subunit B OBD tracks the major groove while additional OBDs encircle the dsDNA as it approaches the collar ring (Figs. 2a, 3 and Supplementary Fig. 3). Second, the 5′ ssDNA is trapped in a groove between collar subunits D and E where the TNS loop of subunit D and Lys310 in the inter-domain linker of subunit E escort the unwound 5′ ssDNA from the complex (Fig. 5). Third, the positions of adjacent OBDs D and E above the collar ring (Supplementary Fig. 3) would prevent the 5′ ssDNA from skipping to an alternative channel at the subunit interfaces. Finally, interactions with the OBD of subunit E further stabilise the path of the 5′ ssDNA. Importantly, our observations of the OBD–RF interactions are consistent with single-molecule fluorescence energy transfer (smFRET) experiments, where DNA unwinding by E1HD alone is significantly less smooth than the process catalysed by E1, where the presence of the E1 OBDs helps to prevent backward slippage on ssDNA and rewinding of duplex DNA[28]. Furthermore, variant proteins with substitutions of residues implicated in DNA interactions demonstrated measurable defects in dsDNA unwinding, while probing in a footprinting assay also suggests that DNA contacts are altered in these variants (Fig. 4, and Supplementary Figs. 5, 6). Therefore, simple strand exclusion mechanisms may be sub-optimal in hexameric helicases, without fork stabilisation and DNA strand escorting mechanisms that enhance the strand separation process.

Interactions with the fork dsDNA have been observed primarily in the monomeric SF1 and SF2 helicases, including RecB of the RecBCD-type helicase-nucleases[41,42], PcrA[9], Hel308[10] and UvrD[43] where they are proposed to have direct mechanical roles in base pair destabilisation in an ATP-dependent power stroke. In contrast, the role of the E1 OBD B-dsDNA interaction in unwinding is indirect, by assisting in the positioning of the RF to prevent reversal of the helicase[28].

Recently obtained structures of the yeast and human CMG helicases show a short stretch of dsDNA entering the complex and a possible exit path for the 3′ ssDNA[17,18], suggesting that it is trapped between specific protein segments and the fork position is also fixed during unwinding. The structure of the yeast CMG helicase (Cdc45, MCM and GINS) bound to the fork protection complex (Csm3/Tof1 and Mrc1) shows Csm3/Tof1 located on the N-terminal tier face of MCM, 'gripping' the dsDNA. While

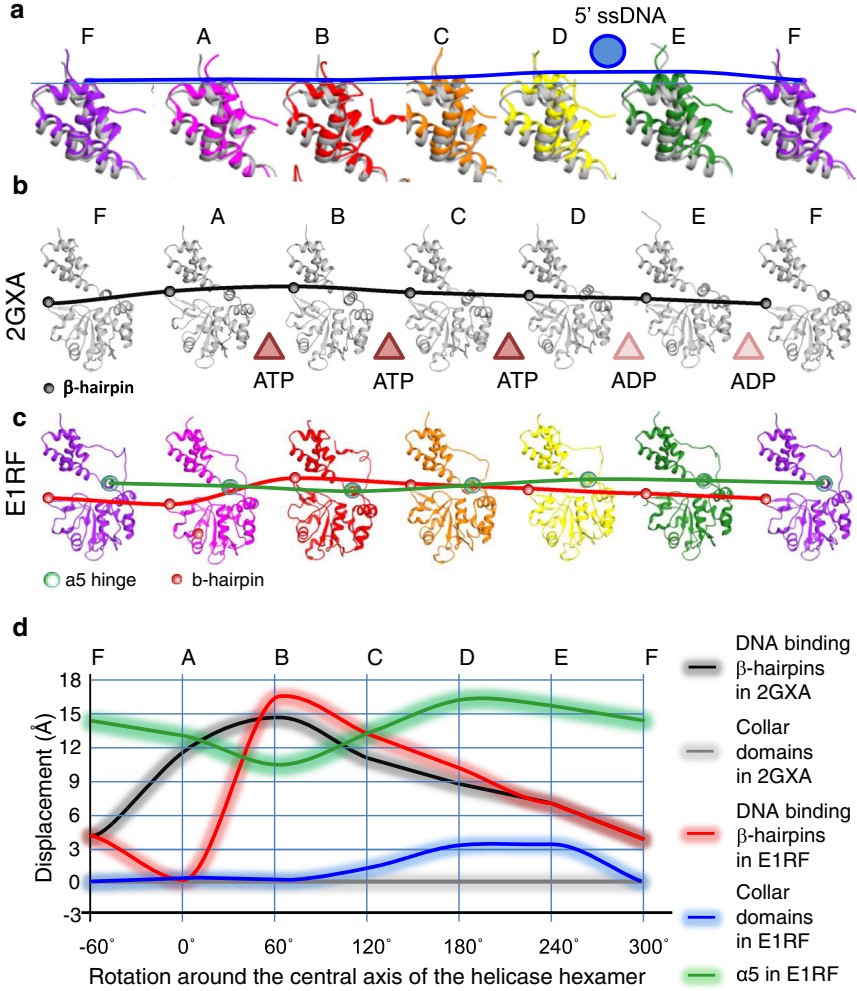

**Fig. 7 Positional variation of the subunit domains in E1RF. a** A tiled representation of the atomic models of the E1RF collar domains (rainbow) overlaid with the corresponding domains of the E1HD/ssDNA/ADP X-ray structure (in grey). The blue curve indicates the collar domain shifts in the EM structure compared to the X-ray structure, which reaches a maximum under the 5′ ssDNA (blue circle). **b** A black curve traces the shifts in the DNA-binding β-hairpins in the X-ray structure; nucleotide states at the subunit interfaces are indicated with brown (ATP) or pale brown (ADP) triangles. **c** Curves tracing the movement of the α5 hinge (green) and DNA binding β-hairpins (red) in E1RF. **d** Graphical representation of the displacements of the collar domains, the β-hairpins and the α-5 hinge point of the E1RF subunits relative to the corresponding subunits in the X-ray structure. The accuracy of the measurements is indicated by the thickness of the curves.

Csm3/Tof1 is required for efficient replication in a reconstituted cell-free system, dsDNA-binding mutants showed no or minimal defects in in vitro DNA replication[18]. Although the phylogenetic similarity between E1 and Mcm2–7 is limited to the AAA+ motor domain, the E1 OBDs may perform a similar function to the fork protection complex. Together, the data suggest that the correct positioning of the dsDNA and relatively lose protein contacts that guide its path are critical for optimal DNA unwinding.

Our cryo-EM E1RF structure revealed the C-terminal acidic tails. They terminate in a groove at the interface between collar domain subunits, with the acidic portion positioned below the 5′ ssDNA. Here, they play an important role in processive unwinding by stabilising the E1 hexameric assembly[32]. Moreover, the position of the electronegative segment, now visualised in E1RF, may also be important for 5′ ssDNA escorting. The exit route of the 5′ ssDNA across the top of the collar domain is predominantly neutral (Supplementary Fig. 5b), while only specific positively charged points (Lys310 in the interdomain linker, and lysines on the surface of OBD E) act to fix the path. The acidic (electronegative) portion of the C-terminal tail of subunit E

may also help to direct the path of the 5′ ssDNA by repulsion. The C-tT is conserved in the related SF3 helicase T-antigen and similar acidic segments are also found in other helicases including the T7 gp4 helicase-primase and TWINKLE[44]. Although the function is likely to be conserved in T-antigen[32], in T7 gp4 the acidic tails are involved in local tethering of the polymerase, which immediately replicates the unwound DNA[40]. Appropriate escorting of all arms of the RF would ensure that the ssDNA strands are separated to prevent re-annealing and facilitate coupling to the DNA replicating apparatus.

To date, only two other hexameric helicases structures, the AAA+-type yeast MCM[17,18] and the RecA-type T7 gp4 helicase-primase[40], have been obtained with the DNA strand separation point (the RFJ) observed at near-atomic resolution. MCM and T7 gp4 both use a planar aromatic residue to stack against a base at the RFJ and mechanically assist in unpairing DNA, although for T7 gp4 the separation pin is provided by the polymerase subunit of the replisome. In E1 base pair separation does not employ a specific functional residue but is by steric exclusion at the entrance to the collar ring. Using the E1RF structure we modelled an entire conformational cycle of the helicase (Fig. 7,

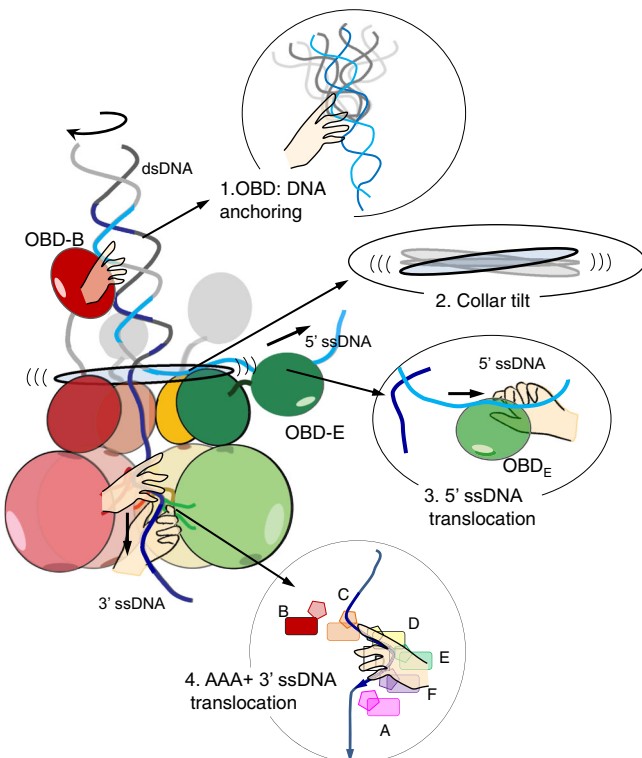

**Fig. 8 Cartoon model for E1 DNA unwinding.** Each arm of the RF DNA associates with specific protein segments indicated with the hands. OBD B tracks the major groove of the inbound, rotating, dsDNA while additional OBDs surround it, stabilising its position above the collar domain. OBD E and residues in the E1HD domain (Asn352) and inter-domain linker (Lys310) chaperone the 5′ ssDNA away from the unwinding point (see Fig. 5). Unwinding is driven by ATP hydrolysis in the AAA+ domain linked to translocation of the 3′ ssDNA. Strand separation could be assisted by a once-per-revolution power-stroke of the collar domain, directly coupled to the ATPase cycle.

Supplementary Movie 1) and this analysis showed that the tilt and elevation of the collar ring follows a wave-like motion around the subunits, providing an auxiliary push against the RFJ. We suggest that base pair separation could be assisted by a once-per-revolution power-stroke directly coupled to the ATPase cycle, allowing E1 to make efficient use of the energy of ATP hydrolysis for translocation coupled DNA unwinding. Although the E1RF structure was determined without ATP or analogues, X-ray structures of the E1HD without[25] and with[14] ssDNA and ADP bound are nearly identical. In particular, the architecture of the nucleotide-binding sites, defined as ATP, ADP and apo type, is maintained and tightly linked to the positions of the DNA-binding hairpins. In E1RF the positions of the β-hairpins correspond well with those in the E1HD/ssDNA/ADP structure (Fig. 6), indicating that the nucleotide-binding site architecture will also be the same and consistent with the coordinated escort model of ssDNA tanslocation[14].

Our data are in full accord with previous structural models[14,25,29], biochemical data[29,32,37] and smFRET observations[28]. The E1 protein participates in the replication process, using both the E1HD and OBD domains for dsDNA *ori* binding, melting[37,38,45] and processive DNA unwinding[28] (Supplementary Movie 2). PV E1 demonstrates how viruses have borrowed functional segments from eukaryotic cells (e.g. the AAA+ domain) and have mimicked the operating principles of the host cell replication initiation apparatus (e.g. the CMG/fork protection complex[18]) to generate a minimalistic but highly streamlined replication machine. Understanding of these viral proteins will help to improve our knowledge of the more complex cellular replication machines and how viruses could be targeted therapeutically when they emerge as threats.

## Methods

**Assembly and analysis of E1 helicase complexes.** Wild-type and variant full-length E1 protein were purified as described previously[38]. Briefly, the protein was expressed as a GST fusion protein and first purified on glutathione sepharose. Following cleavage of the GST tag with thrombin, the protein was purified free from the tag by cation exchange (25 mM sodium posphate pH 7.1, 5 mM DTT, 10% glycerol, 1 mM PMSF, 1 mM EDTA buffer, 50–400 mM NaCl gradient) followed by anion exchange (25 mM Tris–HCl pH 8.4, 5 mM DTT, 10% glycerol, 1 mM PMSF, 1 mM EDTA buffer, 100–400 mM NaCl gradient) chromatography. The E1–RF helicase complex was assembled and purified by gel filtration chromatography (Superdex S200 HR 10/300 GL, GE Healthcare) as previously described[29]. The oligonucleotides 5′-GGCTTGTATTTCACACCGCACCTC AGCGCG(T)$_{20}$ (active strand) and 5′-CCCCCCCCCCGTGCGCGCTGAGGTGCGGTGTGAAATACAAGCC (passive strand) were annealed to generate the RF substrate (30 base pair dsDNA component underlined). Complexes were assembled with 60 μM E1 and 10 μM fork DNA and the gel filtration buffer used was 10 mM Tris–Cl pH 8.0, 225 mM NaCl, 2 mM DTT, 0.1 mM PMSF and 1 mM EDTA. The hexameric peak fractions were concentrated to ~5 mg/ml, snap-frozen in liquid nitrogen and stored at −80 °C for cryo-EM.

Oligonucleotides for helicase assays were 5′ end-labeled with polynucleotide kinase and [γ$^{32}$P]-ATP (7000 Ci/mmol). The substrate used had the same sequence as for RF assembly, given above, or variants with and without the 5′ and 3′ ssDNA arms. Helicase assays[29] with radiolabelled substrates (0.1 nM) were performed in 20 mM HEPES pH 7.2, 135 mM NaCl, 1 mM DTT, 0.1 mg/ml BSA, 0.1% NP40, 3 mM MgCl$_2$, 1 mM ATP. Reactions were incubated for 60 min at 22 °C and terminated by adjusting the reactions to 20 mM EDTA, 0.1% SDS, 10% glycerol, 0.13% w/v bromophenol blue. Product were separated on an 8% poly-acrylamide/ TBE gel containing 0.05% w/v SDS, and gels exposed to phosphor rimager plates (Fujifilm) for imaging and quantification (Fuji FLA3000, image gauge V3.3 software)[29].

For hydroxyl radical footprinting the sequence of the RF substrate was as above. The active strand was 5′ end-labeled (as above) and the passive strand 3′ end-labeled using [α$^{32}$P]-dCTP (3000 Ci/mmol) and Klenow exo- (NEB), followed by a chase with excess unlabelled dCTP. In the latter case, an oligonucleotide lacking the two 3′ C residues was annealed to the active strand to achieve labelling. The substrates were purified by PAGE before assembling 50 μl binding reactions (20 mM Na phosphate pH 7.2, 135 mM NaCl, 0.1% NP40, 0.1 mg ml$^{-1}$ BSA, 1 mM PMSF, 1 mM DTT) with 16 μM E1 proteins and 2.4 μM RF DNA. After 20 min incubation, a 10 μl sample of each was analysed on an agarose gel (TAE running buffer) to confirm complete DNA binding by gel-shift. The remaining reaction was treated with the hydroxyl radical according to the general guidelines of Dixon et al. 1991[46]. Reactions were diluted with an equal volume of 10 mM Tris–Cl pH 8, 0.1 mM EDTA, 100 mM NaCl and extracted twice with an equal volume of phenol/ chloroform/isoamyl alcohol (25:24:1). An equal volume of the reaction was mixed with 98% formamide loading buffer and products resolved on a 15% denaturing urea sequencing gel. Gels were imaged using a phosphor imager (Fuji) and analysed using the lane profiling tool in the image analysis software (Fujifilm, Image Reader V1.8E), generating density traces for the DNA cleavage ladders with peaks proportional to the radioactive signal of the labelled DNA. Wild-type E1RF was compared to variant E1–RF complexes by overlaying the densitometry traces.

ATPase activity was determined in the helicase buffer but with 8.5 mM MgCl$_2$, and 7.5 mM ATP. The released phosphate was determined over time using the charcoal-binding assay of Iggo and Lane[38,47].

**Cryo-EM data collection.** Purified E1RF complex at ~0.05 mg/ml were applied to lacey carbon grids with a continuous carbon support film (EM Sciences). 3 μl of sample was applied and then blotted for 20 s before plunge-freezing the grids and vitrified using a Vitrobot Mark IV (ThermoFisher$^{TM}$) at 100% humidity and 8 °C. Data for the E1RF complex were collected using EPU software (ThermoFisher$^{TM}$) on a Titan Krios electron microscope (ThermoFisher$^{TM}$) operating at 300 kV and equipped with K3 Summit direct electron detector (Gatan Inc.) at the eBIC Diamond light source facility (Harwell, Oxfordshire, UK) and Birkbeck College, London. For the E1RF complex samples, movies (45 frames per movie) were collected with a dose of 1.12 e$^-$/Å$^2$ per frame with a calibrated pixel size of 1.085 Å/ pixel. Images were collected at a range of defoci between −1.2 and −2.5 μm.

**Electron microscopy data processing.** 11,200 movies were aligned using MotionCorr2[48]. CTFFIND4[49] was used to determine defocus values. Micrographs were screened manually to assess CTF quality and selected based on the presence of high-resolution Thon rings at least to 4 Å and beyond for further processing. For particle picking we used crYOLO v1.3.6[50] with the following procedure: a set of 50 randomly selected micrographs were used for manual picking of particles; these selected particle images were used as a model to train the crYOLO particle picking

procedure. This model was optimised by running in several iterations, and tested initially on a sub-set of 100 micrographs for picking ability. The optimised model was then used to pick particles from the entire dataset. RELION 3.0[30] was used to extract selected particle images for the E1RF complex with the box sizes of $300 \times 300$ pixels, the total number was ~560,000 particle images. The extracted particle images were then subjected to two-dimensional (2D) classification in RELION 3.0 and the subset of the images that comprised the best classes, showing secondary structural features, was exported subsequently to cryoSPARC v2.9.0[31]. All following steps in image processing were carried out in cryoSPARC. A set of ~180,000 particle images was selected, based on choosing side views with a few end/tilt views in order to avoid preferred orientation effects on 3D reconstruction. This set of particles was subjected to ab-initio 3D classification implemented in cryoSPARC and running the procedure in multiple rounds, giving six K seeds in each round. 3D maps with clear density for the helicase domain and dsDNA were grouped and used for homogenous 3D refinement, cryoSPARC, using the 3D map of the E1RF complex obtained during the first step of 3D classification. The final 3D map was obtained at a resolution of 3.89 Å at 0.143 FSC threshold (and 4.5 Å at 0.5 FSC threshold). For the fitting, the map was sharpened using option Auto-Sharpen in PHENIX v1.14[51] (Supplementary Figs. 1 and 2).

Local refinements were performed for individual domains of the E1RF complex using masks with soft edges around selected areas of the helicase domains, DNA fork with the collar domains, and OBDs B and E. Small improvements in resolution were observed based on the focused refinement. Later, the overall refined 3D map was used to analyse the OBD flexibility. Focused classification of maps within areas of the OBDs and DNA fork junction was carried out using the 3D variability option in cryoSPARC, based on the usage of three first modes of principal components and generating six clusters. These six maps were analysed for the distribution of densities and results are shown in Supplementary Fig. 3.

**Model building and validation**. Fitting into the final cryo-EM E1RF map was done using as a starting model the X-ray structure of the E1 helicase domain with ssDNA and bound ADP (PDB 2GXA)[14]. Firstly, the correspondence of subunits to the X-ray atomic model was determined by rotating the X-ray structure (rigid body fitting) by ~60°, refinements of the local fitting and assessing the cross-correlation with the EM map. The position with the highest cross-correlation was used as an initial point for the following flexible fit of the hexameric model using normal-mode analysis in iMODFIT v.1.44[35]. Then, the model was refined and validated using PHENIX v1.14 real space refinement[51]. The quality of the model was assessed using COOT v0.8.9.1[52]. The initial model of the DNA fork was built using COOT and its fit into the EM density was refined using the Isolde package[53] and real-space refinement option in PHENIX[51]. Fittings of OBD-B and -E were done using the X-ray structure (PDB 1KSY)[33] as the initial model, fitted as a rigid-body into each OBD block of density within the EM map using Chimera[54,55]. These fits were further locally refined using iMODFIT[35].

A final round of real-space refinement using PHENIX v1.14 with secondary structure restraints was run using the E1RF atomic model based on the independently refined fittings of the helicase domains, the DNA fork and OBD B and E into the cryo-EM map. MOLPROBITY v4.440[56] was used to evaluate the quality of the structures. All data and model statistics are reported in the Supplementary Table.

All figures and movies were produced using UCSF CHIMERA v1.14, CHIMERAX v1[54,55].

**Reporting summary**. Further information on research design is available in the Nature Research Reporting Summary linked to this article.

## Data availability

The E1RF map and atomic model are deposited to the EMDB data base under accession codes EMD-11852 and 7APD [https://doi.org/10.2210/pdb7APD/pdb] correspondingly. Source data are provided with this paper.

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

## Acknowledgements

We acknowledge Diamond Light Source for access to and support of the cryo-EM facilities at the UK national Electron Bio-Imaging Centre (eBIC) (proposal EM14704), funded by the Wellcome Trust, the Medical Research Council (MRC), and the Bio-technology and Biological Sciences Research Council (BBSRC). Part of the Cryo-EM data for this investigation was collected at the ISMB EM facility at Birkbeck College, the University of London with financial support from the Wellcome Trust (202679/Z/16/Z and 206166/Z/17/Z). We thank Y. Chaban with D. Clare (eBIC), and N. Lukoyanova with S. Chen (Birkbeck) for their help with the data collection. D. Houldershaw for computer support in Birkbeck throughout the duration of the project. We thank F. Coscia, Y. Chaban, K. Ryzhenkova for the initial steps in the analysis of this complex and S. Dehghani-Tafti for cloning E1 mutants. This work was supported by BBSRC grants to E.V.O. (BB/R002622/1) and C.M.S. (BB/R001685/1). We thank all reviewers for constructive suggestions during the review of the manuscript, to help improve manuscript clarity.

## Author contributions

E.V.O., and C.M.S. designed the research. J.A.S., B.M., and C.M.S. expressed, purified E1RF, and tested the activity of E1 mutants. A.J. prepared the EM grids, collected EM data. A.J., E.V.O. analysed the data and performed modelling. A.J., E.V.O. and C.M.S. wrote the manuscript; and all authors contributed to and approved the final manuscript.

## Competing interests

The authors declare no competing interests.
