## [Peer Review File · Nature Communications]

Unwinding of a DNA replication fork by a hexameric viral helicaseREVIEWER COMMENTS

Reviewer #1 (Remarks to the Author):

The authors have investigated the structure of a replication fork made up of the Papillomavirus E1 helicase and a forked DNA and tested some of these interactions for their functional importance. Due to the unique features of this homo-hexameric helicase, the authors were able to observe for the first time all DNA strands of a replication fork. In particular the lagging strand has never been observed in the context of a hexameric helicase complex, which shows a sharp bend in the DNA path followed by a transversion along the collar and OBD-E. The structure is of high enough resolution to detail these results and importantly do not require any artificial road blocks, as nucleotides were removed to block DNA unwinding by E1. The authors describe in detail the DNA interactions, although the 4 different interactions are a bit difficult to follow in the figures. The authors present a very nice movie and associated figures that show how specific beta-hairpin engage with the ssDNA and associated structural changes in the ATPase domain. Also, the authors show in extended figure 4 the analysis of the different OBD domains, which were resolved to a lower degree, which is informative. Extended Figure 6 is very nice, explaining one of the DNA interactions in more detail. Have other authors looked at that region in the past? It could be good to systematically present the data on sequence conservation of all DNA contacts (see below). As the purpose of the work is also to inform on the eukaryotic system, it would be great to discuss this in more detail (some more comments below). In general, this is very high-quality manuscript and with the necessary revision, I would be happy to support its publication.

Major points:

- I am a bit concerned about Extended Data Figure 5. It looks identical to Figure 6 in the authors Nucleic Acids Res. 2015 Sep 30; 43(17): 8551–8563 publication, which showed the same footprint. Also, Figure 1 b looks very similar to Figure 2A in the same NAR manuscript – maybe a different exposure. I could be wrong, but it would be good to demonstrate the independence of experiments e.g. show the entire gels. I think this is also in interest of the authors.
- The conformation of the DNA fork in E1RF is maintained by OBD DNA interactions that fix its position for the duration of unwinding, which is reminiscent of the fork protection complex (Yeeles lab LMB) – maybe this can be discussed? How does the smFRET work (citation 27) fit with mutants in the fork protection complex?
- The identification of the four ssDNA interactions is very interesting. For Lys310 it is not said if it is conserved. If so, then it should be mutated to understand its contribution. Also, Lys168 and Lys279 should be mutated to understand their role.
-

Minor comments:

- Abstract: Mention the directionality of the helicase in the beginning.
- Second paragraph discussion – is it possible to link this to figures?
- Unclear - The OBD-RF interactions explain observations from single molecule fluorescence energy transfer (smFRET), where DNA unwinding by E1HD is disrupted with backsliding or rewinding events, while the process catalysed by E1 with the OBDs present is significantly less so (please revise).

A few typos:

Page3, line 39: revise.... "at replication forks" -> generating replication forks that ...

Page 3, line 67: nucleotides of the NA chain interacting -> DNA chain?

Reviewer #2 (Remarks to the Author):

Replicative helicases are core components of replisomes and understanding the unwinding mechanism of these hexameric enzymes has been of great interest to the replication field. In this manuscript, Javed et al. report the 3.9Å structure of the E1 replicative helicase bound to fork DNA in the absence of any nucleotides. This stalled complex shows the 3' end of the ssDNA threading through the central pore of the E1 collar and AAA+ domains, while the 5' ssDNA is diverted on the surface of the hexamer along a groove in between E1 subunits, supporting a strand exclusion

mechanism for DNA unwinding. This work builds on and supports the findings of a large body of prior studies on mechanistic aspects of viral (including E1), bacterial, and eukaryotic replicative helicases from multiple labs. The major novelties of the current work lies in revealing the path of the excluded DNA strand and visualizing a subset of the N-terminal OBD folds, explaining how these domains facilitate DNA unwinding by preventing backtracking of the helicase. The following concerns and questions remain that need be addressed by the authors to substantiate their conclusions.

1) Page 3: "Four of the six known helicase superfamilies (SF3-6) are enzymes assembled from six subunits arranged as hexameric rings, while those of SF1 and 2 are monomeric."
Some SF1/SF2 helicases are dimeric. The authors may want to clarify this point in the introduction.

2) Page 4: "In a low-resolution EM structure of the full-length E1 helicase bound to a synthetic DNA replication fork (RF) the RFJ was mapped at the entrance of the helicase collar domain, suggesting that the unwinding mechanism could not be explained by simple strand exclusion."

The RFJ would be expected to be positioned at the entrance of the helicase collar if a steric exclusion mechanism (which is widely accepted as the mechanism) were used for unwinding, and this is indeed what the authors report in the current manuscript. It is unclear why the authors thought that a steric exclusion could not explain these observations.

3) Page 4: "Here we present a near-atomic resolution (3.9 Å) cryo-EM structure of E1 revealing for the first time the details of dsDNA unwinding by a eukaryotic AAA+ type hexameric helicase."
This is an overstatement as structures of several eukaryotic CMGs bound to fork substrates have already been determined. Although the lagging strand is not visible in these maps, these structures have still revealed many details of the unwinding mechanisms of eukaryotic AAA+ replicative helicases.

4) Ext. Data Fig. 2a: The authors should include an FSC curve comparing pdb model versus map.

5) Ext. Data Fig. 3: It would be helpful if the authors used a consistent color scheme in a and b. Helices and strands could be differentiated by different geometric shapes in b.

6) Fig. 2: Side chain densities for lysine 310-E and asparagine 352-D should be shown in a zoom-in view considering the critical role the authors think these residues play in the unwinding mechanisms

7) Docking of the OBD structure into the cryo-EM map indicates that the OBD-DBL interacts with the major groove of the dsDNA segment. What effect do mutations in the DBL have on E1 helicase activity? Related to this issue, how do mutations in the DBL and deletions of the OBD alter the footprinting pattern in Ext. Data Fig. 5? Testing these protein constructs in these assays is critical for validating the cryo-EM structure, which is only resolved to 3.9 Å.

8) The authors mention the DBH as a second DNA binding region, but it is not clear where this region is located in the structure. The authors should clarify this issue.

9) The authors highlight the TNS loop and lysine 310, which interacts with the 5' ssDNA. Although the authors perform mutational analysis of N352 in the TNS loop, lysine 310 is not analyzed. What effect do mutations of lysine 310 have on unwinding activity? Is lysine 310 conserved in related replicative helicases? Do mutations in the TNS loop and lysine 310 alter the footprinting pattern (Ext. Data Fig. 5)?

10) It would be helpful if the authors added a movie of the entire structure, also showing cryo-EM density of key structural features.

11) I strongly encourage the authors to use commas within sentences to improve clarity.

Minor points:

- 1) Page 3: NA is not defined.
- 2) Fig. 4 is referenced before Fig. 3. The authors should switch the order of these figures.

Reviewer #3 (Remarks to the Author):

Javed, Major, Stead et al. describe here the structure of the bovine papillomavirus helicase E1 in complex with a DNA fork. The structure is interesting and highlights the mechanisms of interaction of E1 to both dsDNA and ssDNA, as well as the molecular basis for strand separation. It is interesting to have a structure of E1RF as an example of a minimal system that shows how DNA unwinding goes on while the complex is maintained (possible basis for explaining processivity). Unfortunately, the paper could be much better articulated and compared with the existing literature. The paper is almost entirely based on one structure that should be therefore presented more thoroughly than it currently is. Upon the corrections/additions suggested, the paper could be considered for publication.

Essential revisions:

- 1) The collar position implication in strand separation is presented as hard evidence, while it is rather suggested by the structural analysis. The authors should provide a final schematic/cartoon where they clarify their model for strand separation. The suggested idea for strand separation seems very plausible, but it should be presented as a suggested model not as hard evidence. For hard evidence, other type of non-structural experiments would be required. It is understandable that a paper cannot include everything and be completely conclusive.
- 2) The general description of the complex structure, first paragraph of Results, should be re-written to make it easier to guide the reader through the map. For example, in line 112 there is a reference to Figure 1d about visualization of E1HD, but then this information is not found in the figure. In general, I find the description of the complex (line 92-132) very confusing. When mentioning the ssDNA-binding hairpins it is better to stick to those and complete their description /comparison instead of introducing the description of old NS data and of the, presumably, OBDs.
- 3) It is very important that the authors provide more detailed information both in the text and the Methods about the structure and the way certain conclusions were obtained. - What is the resolution of the 2 visible OBDs? Has local refinement been attempted? If so, how? What is the resolution of the b-hairpins? Any comment also on the ATP-binding pockets? The image-processing session should be more detailed. For example, based on line 569, it is not clear whether 1 or 2 maps were finally generated, at 4.5 or at 3.9Å resolution? ED Figure 1 does not help either. The depiction of the local resolution looks strange: maybe other range values should be chosen? From this it looks like most of the map was at 3.5Å which it is unlikely to be, looking at the map itself. Also, from the main figure I would expect the OBDs to be at lower resolution overall. This does not seem to be the case from the look of this depiction. How was this local resolution map values generated? Do you have a locally filtered map? Did you try local sharpening with other methods than Phenix? Did the author try Nautilus or other programs for automatic building of the ssDNA stretches? How was the FSC curve generated? Maybe it is just as good to report the FSC curve as it comes out of cryoSPARC or/and Phenix. ED Figure 4 presents extensive 3D classification. How were the different classes obtained? Did the author try to do focused classification? How? Using what masks? This part is very interesting and should be described better and discussed a bit more in the text. A general suggestion for some figure: the authors could show also the map filtered according to resolution or simply filtered at low resolution to better make the point of the location of OBD E and F
- 4) The interesting aspects of this paper are the interaction of the OBD with the dsDNA and the ssDNA, so why is a lot of this information relegated to Extended Data e.g. ED figure 5 or 6?
- 5) The authors speculate quite a bit on the alpha5 hinge, while it actually looks like this hinge hardly moves. Maybe this is more evident once one looks at the actual structure and model, which

would be nice to have access to in a second round of review.

6) In line 287 the authors state that "In E1 base pair separation does not employ a specific functional residue but is by steric exclusion at the entrance to the collar ring. The E1RF structure demonstrates that separation is assisted by a once-per-revolution powers stroke directly coupled to the ATPase cycle, as E1 makes optimal use of the energy of ATP hydrolysis for translocation coupled DNA unwinding." This is hardly proven by the data. Instead, it seems that residues Lys310 and Asn 352 are extremely important. The lack of specific information about the ATP-binding pockets makes it hard to correlate, even though comparison to the crystal structure suggests so. Should be explained better in the discussions for the sake of correctness.

7) Check figure numbering! Figure 4 is mentioned before figure 3.

Minor revisions:

1) The abstract should be clearer. Please re-phrase "Here we present the first cryo-EM structure of a AAA+ helicase, E1," with something like "Here we present cryo-EM structure of the AAA+ helicase E1 in complex with ...". The author can somewhere else specify that it is the first structure of E1 in complex with a replication fork where both ssDNA "arms" are visible; but as it is written now it looks as if they had the very first structure of E1 in complex with DNA which is not true.

2) In general, the abstract is not very clear to me.

3) Line 132, the residues of the C-tT: was it present in the construct but not visible in the X-ray density?

4) Line 80-81 again a claim of first time – not quite correct. It is true that it is the first time that the lagging strand is so well visible and interacting with a OBD, but not it is "the first time that resolution (3.9 Å) cryo-EM structure of E1 revealing for the first time the details of dsDNA unwinding by a eukaryotic AAA+ type hexameric helicase". The way of transferring this type of information is at the end responsibility of the editor, I believe.

5) I find the constant reference to the previous negative-stain structure a bit confusing. Ok to comment about it in the discussions, but what role does the comment about NS play in line 143?

6) Line 193 "show measurable defects". How measurable? 2times? 10 times?

7) Line 198-204. Could it be rephrased. Hard to follow which OBD is what.

8) Line 218-219: So is the b-hairpin of B contacting the DNA? If one writes while in the A in not contacting, one expects that in B it is contacting. One can look at Figure 3 that where B seems non-contacting. Adjust the text.

9) Line 240-241: As the six subunits of the E1 hexamer each complete a 239 cycle of ATP hydrolysis pulling six nucleotides through the AAA+ motor there is an integral 240 power stroke equivalent in length to the displacement of one base pair (~3 Å). Is there biochemical evidence for it?

10) Scale bar on EDFigure1a.

Answers to the reviewers comments

We thank all reviewers for their constructive and positive comments. We have addressed all raised points below. The reviewers' comments are in blue and our answers are in black. Unfortunately, to perform experiments requested by the reviewers took much more time than anticipated due to the Covid pandemic and Brexit. Deliveries of chemicals necessary for these experiments were dramatically delayed and this has delayed the resubmission of the manuscript.

Reviewer #1 (Remarks to the Author):

... Due to the unique features of this homo-hexameric helicase, the authors were able to observe for the first time all DNA strands of a replication fork. In particular the lagging strand has never been observed in the context of a hexameric helicase complex, which shows a sharp bend in the DNA path followed by a transversion along the collar and OBD-E. The structure is of high enough resolution to detail these results and importantly do not require any artificial road blocks, as nucleotides were removed to block DNA unwinding by E1. The authors describe in detail the DNA interactions, although the 4 different interactions are a bit difficult to follow in the figures. The authors present a very nice movie and associated figures that show how specific beta-hairpin engage with the ssDNA and associated structural changes in the ATPase domain. Also, the authors show in extended figure 4 the analysis of the different OBD domains, which were resolved to a lower degree, which is informative. Extended Figure 6 is very nice, explaining one of the DNA interactions in more detail. Have other authors looked at that region in the past? It could be good to systematically present the data on sequence conservation of all DNA contacts (see below). As the purpose of the work is also to inform on the eukaryotic system, it would be great to discuss this in more detail (some more comments below). In general, this is very high-quality manuscript and with the necessary revision, I would be happy to support its publication.

We thank the reviewer for the positive assessments of our work and we have endeavoured to address all issues raised by this and other reviewers. The figures were modified to make our results clearer for readers. We are happy that figure 6 (now Extended Data Fig. 5) was informative and it is now slightly extended, providing more information on the role of new variant proteins in dsDNA unwinding that we have analysed. Residues in the collar domain tunnel have been examined before (K356 and K359) and the reference to the study was given (main text ref. 37, revised version).

Figure 1 b looks very similar to Figure 2A in the same NAR manuscript – may be a different exposure

Figure 1b has been replaced with an experimental series employing the fork substrate used to determine the structure of E1RF, along with control substrates with 5' or 3' arms only (now Fig.1d). The only difference to the substrate used in the NAR report is a slightly longer passive strand (3 nucleotides). Two of the additional nucleotides were larger guanine residues placed near the fork junction (rather than all poly dC) in anticipation that it would help in EM visualisation. The fundamental

observation that E1 preferential unwinds the fork substrate was reported in detail in the NAR manuscript.

I am a bit concerned about Extended Data Figure 5

The figure has been replaced with a new one incorporating new experiments that were requested. This took more time than had been anticipated. The new data for variant proteins compared to wild-type provide further support for our model and the interactions observed with the arms of the fork.

As the purpose of the work is also to inform on the eukaryotic system, it would be great to discuss this in more detail (some more comments below). In general, this is very high-quality manuscript and with the necessary revision, I would be happy to support its publication.

We thank the reviewer for this advice. We have extended the discussion taking in to account more published data in the eukaryotic systems.

The conformation of the DNA fork in E1RF is maintained by OBD DNA interactions that fix its position for the duration of unwinding, which is reminiscent of the fork protection complex (Yeeles lab LMB) – maybe this can be discussed? How does the smFRET work (citation 27) fit with mutants in the fork protection complex?

This is a good point. There are indeed some similarities with the fork protection complex. This has been discussed along with the parallels with the mutants analysed by the Yeeles' lab. It has been shown by the Joshua-Tor and Ha labs (2014, ref 27 now 28) that E1 complexes without the OBD and N-terminus demonstrated the following: "The majority of traces showed highly diverse nonmonotonic unwinding patterns characterized by repetitive increases and decreases in FRET consistent with rewinding (or slippage) and unwinding movements." We have now analysed E1 variants, targeting residues identified as functional in our structural data, with amino acid substitutions in the OBD and OBD-E1HD inter-domain linker and they show defects in DNA binding and unwinding that likely explain more clearly the basis of these previous observations (Fig. 4, 7 and Extended Data Fig. 5).

The identification of the four ssDNA interactions is very interesting. For Lys310 it is not said if it is conserved. If so, then it should be mutated to understand its contribution. Also, Lys168 and Lys279 should be mutated to understand their role.

We have considered this question ourselves and have now provided a new Extended Data Figure 4 with a sequence alignment using representatives of all the major papillomavirus genera. Lys310 in the inter-domain linker is not conserved; we should have mentioned this and it is now said in the manuscript. However, Lys168 and Lys279 (OBD E interaction with the 5' passive ssDNA) are conserved or substituted with arginine, as is the case for Lys183 and 186 (OBD B interaction with the dsDNA).

In helicase assays we have now analysed a double variant protein K183A/K186A, K310A, K279A and K168A single variants and a triple variant K168A/K310A/N352G. We have also probed E1-DNA interactions using the hydroxyl radical footprinting

assay for E1 K183A/K186A (to probe the dsDNA interaction) and E1 K168A/K310A/N352G (to probe the interaction with the 5' passive ssDNA), see the answer to reviewer 2, below. The text was amended accordingly, reporting these new data.

Minor comments:

Abstract: Mention the directionality of the helicase in the beginning.

The directionality is now mentioned at the beginning.

Second paragraph discussion – is it possible to link this to figures?

Thank you for this practical recommendation, we have linked the text to Figure 2a and extended data Figure 3. The text has been amended accordingly. In general we have linked the discussion to the figures more clearly.

Unclear - The OBD-RF interactions explain observations from single molecule fluorescence energy transfer (smFRET), where DNA unwinding by E1HD is disrupted with backsliding or rewinding events, while the process catalysed by E1 with the OBDs present is significantly less so (please revise).

We thank the reviewer for the suggestion to help improve the readability of the manuscript. The sentence has been modified to improve clarity:

“Importantly, our observations of the OBD-RF interactions are consistent with single molecule fluorescence energy transfer (smFRET) experiments, where DNA unwinding by E1HD alone is significantly less smooth than the process catalysed by E1, where the presence of the E1 OBDs helps to prevent backward slippage on ssDNA and rewinding of duplex DNA²⁸.”

A few typos:

Page3, line 39: revise.... “at replication forks” -> generating replication forks that ...

We thank the reviewer for this suggestion, the correction was done.

Page 3, line 67: nucleotides of the NA chain interacting -> DNA chain?

It should be NA chain because Rho is an RNA helicase, so we are talking about RNA and DNA helicases together.

Reviewer #2 (Remarks to the Author):

Replicative helicases are core components of replisomes and understanding the unwinding mechanism of these hexameric enzymes has been of great interest to the replication field. In this manuscript, Javed et al. report the 3.9Å structure of the E1 replicative helicase bound to fork DNA in the absence of any nucleotides. This stalled complex shows the 3' end of the ssDNA threading through the central pore of the E1 collar and AAA+ domains, while the 5' ssDNA is diverted on the surface of the hexamer along a groove in between E1 subunits, supporting a strand exclusion mechanism for DNA unwinding. This work builds on and supports the findings of a

large body of prior studies on mechanistic aspects of viral (including E1), bacterial, and eukaryotic replicative helicases from multiple labs. The major novelties of the current work lies in revealing the path of the excluded DNA strand and visualizing a subset of the N-terminal OBD folds, explaining how these domains facilitate DNA unwinding by preventing backtracking of the helicase. The following concerns and questions remain that need be addressed by the authors to substantiate their conclusions.

We thank the reviewer for the positive assessment of our MS and acknowledgement of the novelty of our work. See below our answers to the reviewer.

1) Page 3: “Four of the six known helicase superfamilies (SF3-6) are enzymes assembled from six subunits arranged as hexameric rings, while those of SF1 and 2 are monomeric.” Some SF1/SF2 helicases are dimeric. The authors may want to clarify this point in the introduction.

We thank the reviewer for this suggestion. Indeed, it is an ongoing, contentious, issue as to whether dimerization is obligatory for some of these helicases to function. The text has been modified by the addition of two citations:

“Four of the six known helicase superfamilies (SF3-6) are enzymes assembled from six subunits arranged as hexameric rings, while those of SF1 and 2 are monomeric but sometimes function as dimers^{2,3}.”

2) Page 4: “In a low-resolution EM structure of the full-length E1 helicase bound to a synthetic DNA replication fork (RF) the RFJ was mapped at the entrance of the helicase collar domain, suggesting that the unwinding mechanism could not be explained by simple strand exclusion.”

The RFJ would be expected to be positioned at the entrance of the helicase collar if a steric exclusion mechanism (which is widely accepted as the mechanism) were used for unwinding, and this is indeed what the authors report in the current manuscript. It is unclear why the authors thought that a steric exclusion could not explain these observations.

We agree with the reviewer; the original wording was not clear. It is now replaced with:

“In a low-resolution EM structure of the full-length E1 helicase bound to a synthetic DNA replication fork (RF) the unwinding point was mapped at the entrance of the helicase collar domain²⁹, consistent with the steric exclusion model. However, the structure also showed extensive interactions of the N-terminal domains of the protein with the DNA ahead of the replication fork.”

3) Page 4: “Here we present a near-atomic resolution (3.9 Å) cryo-EM structure of E1 revealing for the first time the details of dsDNA unwinding by a eukaryotic AAA+ type hexameric helicase.”

This is an overstatement as structures of several eukaryotic CMGs bound to fork substrates have already been determined. Although the lagging strand is not visible in these maps, these structures have still revealed many details of the unwinding mechanisms of eukaryotic AAA+ replicative helicases.

Our data do reveal clearly for the first time all arms of the RF, so we disagree that this part is an overstatement. However, we agree that previous (MCM) structures without the lagging strand visible in the maps have provided many important mechanistic insights. We have modified the final paragraph of the introduction accordingly.

4) Ext. Data Fig. 2a: The authors should include an FSC curve comparing pdb model versus map.

We thank the reviewer for the comment. The figure was amended and the FSC between the fitted atomic model and the EM density map has been added (Extended Data Fig. 2a).

5) Ext. Data Fig. 3: It would be helpful if the authors used a consistent color scheme in a and b. Helices and strands could be differentiated by different geometric shapes in b.

Due to the many amendments in the manuscript and addition of new figures, we have decided that this figure is no longer informative so it has been removed from the manuscript. Information on secondary structural elements is provided now in Ext. Data Fig. 4.

6) Fig. 2: Side chain densities for lysine 310-E and asparagine 352-D should be shown in a zoom-in view considering the critical role the authors think these residues play in the unwinding mechanisms

We thank the reviewer for this advice and this has been amended in figure 5b&c.

7) Docking of the OBD structure into the cryo-EM map indicates that the OBD-DBL interacts with the major groove of the dsDNA segment. What effect do mutations in the DBL have on E1 helicase activity? Related to this issue, how do mutations in the DBL and deletions of the OBD alter the footprinting pattern in Ext. Data Fig. 5? Testing these protein constructs in these assays is critical for validating the cryo-EM structure, which is only resolved to 3.9 Å.

As mentioned in the response to reviewer 1 above, we have constructed additional mutants and assayed helicase activity of purified variant proteins as the reviewer asked. All mutants show defects in helicase activity, consistent with the structural data (Figure 4 and Extended Data Figure 5). Probing the interactions in the footprinting assay is challenging for several reasons. First, the homo-hexameric nature of the helicase does not allow us to target a single subunit in the complex. Second, the channelling of the paths of the fork arms is achieved by relatively prominent structural features (*i.e.* the trapping of the 5' passive ssDNA and the dsDNA is achieved largely by the positioning of the OBDs as a whole in their elevated position in the complex (Extended data Figure 3). Third, the interactions are expected to be relatively loose by nature and the combination of many weak interactions along the channelled paths. Significantly, a report from the Yeeles' lab (*Mol. Cell* **78**, 926-940 (2020)) demonstrated no significant effects of amino acid substitution in the dsDNA interacting segments of the fork protection complex (see response to reviewer 1 and amended paragraph in the discussion). In our case, the

defects in helicase activity observed for E1 variants were small but reproducible, so we decided to combine appropriate substitution in individual proteins for footprinting analysis. We performed hydroxyl radical footprinting of E1RF complexes using a double mutant K183A/K186A, to probe the interaction with the dsDNA, and a triple mutant K168A/K310A/N352G to probe interactions with the 5' passive ssDNA strand. The double mutant showed increased susceptibility to OH• cleavage (decreased protection) in the dsDNA on both strands while the triple mutant did not, as would be predicted from our structural data. The triple mutant showed a subtle decrease in susceptibility to OH• cleavage in the 5' passive ssDNA. This is consistent with a role for the amino acids in providing relatively loose, guiding, contacts with the ssDNA, for transit away from the complex.

8) The authors mention the DBH as a second DNA binding region, but it is not clear where this region is located in the structure. The authors should clarify this issue.

We thank the reviewer for the comment. To make that point clear we have modified figure 3 and indicated the positions of the DNA DBH and DBL. The text has been amended accordingly.

9) The authors highlight the TNS loop and lysine 310, which interacts with the 5' ssDNA. Although the authors perform mutational analysis of N352 in the TNS loop, lysine 310 is not analyzed. What effect do mutations of lysine 310 have on unwinding activity? Is lysine 310 conserved in related replicative helicases? Do mutations in the TNS loop and lysine 310 alter the footprinting pattern (Ext. Data Fig. 5)?

Please see the response to reviewer 1 and above to point 7. The homology between E1 and MCM is limited to the AAA+ region.

10) It would be helpful if the authors added a movie of the entire structure, also showing cryo-EM density of key structural features.

We need to apologise, but we had some difficulties in preparation of such a movie. It proved exceptionally challenging and we believe the task is far beyond the scope of non-professional movie makers.

11) I strongly encourage the authors to use commas within sentences to improve clarity.

We thank the reviewer for the advice and made the amendments accordingly.

Minor points:

1) Page 3: NA is not defined.

The abbreviation (nucleic acid) was defined: "...short single-stranded nucleic acid (NA) segments".

2) Fig. 4 is referenced before Fig. 3. The authors should switch the order of these figures.

We thank the reviewer for this comment... We have checked the text and made the appropriate amendments (the figures were changed globally; we believe the numbering is sequential).

Reviewer #3 (Remarks to the Author):

The structure is interesting and highlights the mechanisms of interaction of E1 to both dsDNA and ssDNA, as well as the molecular basis for strand separation. It is interesting to have a structure of E1RF as an example of a minimal system that shows how DNA unwinding goes on while the complex is maintained (possible basis for explaining processivity). Unfortunately, the paper could be much better articulated and compared with the existing literature. The paper is almost entirely based on one structure that should be therefore presented more thoroughly than it currently is. Upon the corrections/additions suggested, the paper could be considered for publication.

We thank the reviewer for the good advice on how to improve our manuscript. We are happy that the reviewer revealed a vivid interest in our results and probed the details of the processing.

Essential revisions:

1) The collar position implication in strand separation is presented as hard evidence, while it is rather suggested by the structural analysis. The authors should provide a final schematic/cartoon where they clarify their model for strand separation. The suggested idea for strand separation seems very plausible, but it should be presented as a suggested model not as hard evidence. For hard evidence, other type of non-structural experiments would be required. It is understandable that a paper cannot include everything and be completely conclusive.

We agree and thank the reviewer for the advice and practical considerations. We have reported the data as suggestive rather than hard evidence, where appropriate in the description of the results and discussion. We have also provided a new figure (Extended Data Fig. 6) to support our observations and ideas. We agree that detailed probing by more complex techniques would be required to provide further evidence, but this is for future study. We have provided a final schematic to clarify our model (Fig. 9).

2) The general description of the complex structure, first paragraph of Results, should be re-written to make it easier to guide the reader through the map. For example, in line 112 there is a reference to Figure 1d about visualization of E1HD, but then this information is not found in the figure. In general, I find the description of the complex (line 92-132) very confusing. When mentioning the ssDNA-binding hairpins it is better to stick to those and complete their description /comparison instead of introducing the description of old NS data and of the, presumably, OBDs.

We have endeavoured to make clarifying amendments, in the text and figures, in accordance with the comment. The diffuse density visible above of the E1HD region in class averages suggests a flexible arrangement of the OBD domains (now Ext.

Data Figs. 1a and 3). There is a rod of density (~20 Å in width) attached to the upper tier and slightly tilted with respect to central axis of E1HD (17°) which is labelled now (now Ext. Data Fig. 1a). The text related to the 3' ssDNA interactions with the AAA domains has been modified accordingly.

3)

a . What is the resolution of the 2 visible OBDs? Has local refinement been attempted?

The resolution of the visible OBDs was around 4.5-7 Å, a local refinement was performed and is described in the methods: focused classification in cryoSPARC.

b. What is the resolution of the β -hairpins?

The resolution of the β -hairpins was ~ 3.5 Å; see Figure 6a and Extended Data Fig. 2b, the side chains are well identifiable.

c. Any comment also on the ATP-binding pockets?

ATP was not used in our experiments for complex formation, so it was impossible to say anything precise in relation to nucleotide binding in our structure. However, the alignment of E1RF with the X-ray structures (with and without ADP) was based on the DNA binding β -hairpins and the position of the hairpins in the ssDNA binding tunnel is directly related to the configuration of the nucleotide binding sites at the subunit interfaces. Accordingly, the nucleotide binding state can be inferred. In the future we plan to extend our research and determine structures with bound nucleotides (ADP and ATP analogues).

d. The image-processing session should be more detailed. For example, based on line 569, it is not clear whether 1 or 2 maps were finally generated, at 4.5 or at 3.9Å resolution?

We regret that some details in our description have not appeared obvious to the reviewer. There is a lot of confusion related to the subject of how to assess the resolution of a map obtained. There are TWO options how to assess the resolution: to use a threshold 0.5 (van Heel and Frank) or to use the threshold 0.143 (Henderson). We have indicated BOTH options: 4.5 Å resolution at the threshold of 0.5, and the same map has an average resolution of 3.9 Å at the threshold of 0.143 (see Methods, electron microscopy data processing, end of the first paragraph). We have amended the text to make it clearer to a reader and within the main text of the manuscript we have used the criterion suggested by Henderson.

e. The depiction of the local resolution looks strange: maybe other range values should be chosen? From this it looks like most of the map was at 3.5Å which it is unlikely to be, looking at the map itself. Also, from the main figure I would expect the OBDs to be at lower resolution overall.

The reviewer has raised a good point. We have changed the range of the resolution at which the assessment was done and now it is correct. The central part of the complex was resolved at a resolution of ~ 3.5 Å. We were able to see main side chains. However, the periphery was at a lower resolution, specifically for the OBDs. See the new figure, Ext. Data Fig. 1.

f. Did you try local sharpening with other methods than Phenix?

We have used only “AutoSharpen” in PHENIX v1.14.

g. Did the author try Nautilus or other programs for automatic building of the ssDNA stretches?

No. We have used Isolde package, the reference has now been added.

h. How was the FSC curve generated?

In cryoSPARC, see methods.

i. Figure 4 presents extensive 3D classification. How were the different classes obtained? Did the author try to do focused classification?

We think the reviewer means the former extended data figure 4 (now Extended Data Fig. 3). However, as we described in the methods, we performed a focused classification in cryoSPARC. This part of the Methods has now been extended.

j. A general suggestion for some figure: the authors could show also the map filtered according to resolution or simply filtered at low resolution to better make the point of the location of OBD E and F

Filtering was not considered necessary since at the classification stage with low numbers of images the resolution was low. However, this low resolution enabled us to see the location of the flexible OBDs.

4) The interesting aspects of this paper are the interaction of the OBD with the dsDNA and the ssDNA, so why is a lot of this information relegated to Extended Data e.g. ED figure 5 or 6?

We thank the reviewer for the suggestion. We have now provided new data in the main manuscript on mutational analysis of E1, to test the hypothesis that protein-DNA interactions revealed by the structural data are important for DNA unwinding. Please see the response to reviewer 1 and 2 above. We have left the analysis Asn352 where we analysed multiple substitutions of this single residue in the Extended data. We believe this represents a good balance.

5) The authors speculate quite a bit on the alpha5 hinge, while it actually looks like this hinge hardly moves. Maybe this is more evident once one looks at the actual structure and model, which would be nice to have access to in a second round of review.

We would like to focus the reviewer’s attention to Figure 8c&d (former Figure 4c). We would argue that the ~ 6 Å movements in the alpha 5 shifts are hardly insignificant.

6) In line 287 the authors state that "In E1 base pair separation does not employ a specific functional residue but is by steric exclusion at the entrance to the collar ring. The E1RF structure demonstrates that separation is assisted by a once-per-revolution powers stroke directly coupled to the ATPase cycle, as E1 makes optimal use of the energy of ATP hydrolysis for translocation coupled DNA unwinding." This is hardly proven by the data. Instead, it seems that residues Lys310 and Asn 352 are extremely important. The lack of specific information about the ATP-binding pockets makes it hard to correlate, even though comparison to the crystal structure suggests so. Should be explained better in the discussions for the sake of correctness.

We thank the reviewers for the points highlighted. We would like to affirm again that we draw implications based on our structural observations that would require further biochemical/biophysical/structural validation. The implication of the conformational wave propagating around the ring are that the collar domain would implement a once per revolution power stroke. This same principal assumption of a propagating conformational wave underpins all mechanistic models of NA translocation by hexameric helicase. Accordingly, we believe our proposition is valid, but where it is described and discussed we have acknowledged that the structural observations allow us to propose a model.

We agree that information on the ATP binding pockets would be useful to have. However, although the E1RF structure was determined without specific information on the ATP binding pockets, X-ray structures of the E1HD without and with ssDNA and nucleotides bound are nearly identical (PDB 2GXA for nucleotide bound E1HD and PDB 2V9P for non-nucleotide/ssDNA bound E1HD). In particular, the architecture of the nucleotide binding site, defined as ATP, ADP and apo type, is maintained and tightly linked to the relative position of the DNA binding hairpins, that were defined in E1RF and are consistent with the coordinated escort model of ssDNA translocation.

The discussion in the main text has been updated accordingly.

7) Check figure numbering! Figure 4 is mentioned before figure 3.

Thank you for your attentive reading. Figure numbering was changed in the manuscript due to the modifications made.

Minor revisions:

1) The abstract should be clearer. Please re-phrase "Here we present the first cryo-EM structure of a AAA+ helicase, E1," with something like "Here we present cryo-EM structure of the AAA+ helicase E1 in complex with ...". The author can somewhere else specify that it is the first structure of E1 in complex with a replication fork where both ssDNA "arms" are visible; but as it is written now it looks as if they had the very first structure of E1 in complex with DNA which is not true.

Yes, we agree with the reviewer and we have clarified this. Please also see the response to reviewer 1.

2) In general, the abstract is not very clear to me.

In the absence of specific comments, we were not sure what was unclear. Modifications have been made in accordance with the suggestions of other reviewers.

3) Line 132, the residues of the C-tT: was it present in the construct but not visible in the X-ray density?

The residues of the C-tT were not present in the construct used by Joshua-Tor et al. (*Nature*. 2006; 442:270–275) to determine the structure of E1HD with ssDNA and ADP bound. The C-tTs were in the construct used to determine the structure without nucleotides and DNA bound (Sanders et al. *Nucleic Acids Res.* 2007; 35:6451–6457). However, only a few residues of the C-tT were visible in the X-ray density. This has been clarified in the manuscript. The cryo EM structure has revealed a significantly longer chain of the C-tT, except only the last 8 amino acids.

4) Line 80-81 again a claim of first time – not quite correct. It is true that it is the first time that the lagging strand is so well visible and interacting with a OBD, but not it is “the first time that resolution (3.9 Å) cryo-EM structure of E1 revealing for the first time the details of dsDNA unwinding by a eukaryotic AAA+ type hexameric helicase”. The way of transferring this type of information is at the end responsibility of the editor, I believe.

We agree that the wording was misleading. Previously obtained EM structures did not reveal the lagging strand, while they have provided important mechanistic insights. We have acknowledged this in the revised manuscript.

5) I find the constant reference to the previous negative-stain structure a bit confusing. Ok to comment about it in the discussions, but what role does the comment about NS play in line 143?

The text has been modified according to the comments of the reviewers.

6) Line 193 “show measurable defects”. How measurable? 2times? 10 times?

We agree that this should have been stated. In the revised manuscript a quantitative measure of the defects has been defined in the text for all mutants, while the quantified data is also presented in graphs.

7) Line 198-204. Could it be rephrased. Hard to follow which OBD is what.

It has been mentioned from which subunit the given OBD came from. Now the text has been amended and the OBDs are more clearly labelled.

8) Line 218-219: So is the b-hairpin of B contacting the DNA? If one writes while in the A is not contacting, one expects that in B it is contacting. One can look at Figure 3 that where B seems non-contacting. Adjust the text.

Yes, we agree: the β -hairpin of A does not contact the 3' ssDNA. The β -hairpin of subunit B does not make direct contact with the 3' ssDNA either, since it's His507 is

turned away from ssDNA compared to the crystal structure (Fig. 6). The text has been amended.

9) Line 240-241: As the six subunits of the E1 hexamer each complete a cycle of ATP hydrolysis pulling six nucleotides through the AAA+ motor there is an integral power stroke equivalent in length to the displacement of one base pair (~3 Å). Is there biochemical evidence for it?

Please see the answer to point 6 above, major comments.

10) Scale bar on ED Figure1a.

Thanks, done.

REVIEWERS' COMMENTS

Reviewer #1 (Remarks to the Author):

The manuscript is much improved. I would have liked a marked up version - without this it was difficult to know what changed. The discussion is more inclusive and more informative. Once the comments below are addressed, I would support publication:

Abstract "The 3' single stranded DNA is pulled through the helicase by the rotational height-adjusted movement of the AAA+ DNA-binding β -hairpins". The section in bold is correct, but not easy to understand for the standard reader. This should be made more accessible.

Abstract / discussion "while a cyclical movement of collar domain subunits optimises the use of energy from nucleotide hydrolysis for strand separation". I am not sure that this has been demonstrated. Has the energy been calculated to demonstrate that? If it is a speculative model, then this should be indicated.

Introduction "In cells the principal replication helicase". Usually it is called replicative helicase not replication helicase.

Figure 7 – Why did the authors use a low concentration of E1. In their recent NAR publication, they observed clear protection of the RFJ only at higher protein concentrations. Overall the protective effect appears rather minor, so maybe this is better placed in the supplementary figures. It is likely that Hydroxy-radical footprint is not ideal to detect these rather weak and transient interactions, or that additional amino acids participate in this process

Reviewer #2 (Remarks to the Author):

In this revised manuscript, the authors address some but not all concerns from the original submission. The following concerns remain or arise from the revision:

Page 4:

"Here we present a near-atomic resolution (3.9 Å) cryo-EM structure of E1 revealing clearly for the first time all arms of the RF and the dsDNA unwinding point for a eukaryotic AAA+ hexameric helicase."

This sentence remains misleading. Although the E1 structure may have implications for the unwinding mechanism employed by eukaryotic AAA+ helicases, E1 itself is a viral helicase and not a eukaryotic one as this sentence implies.

Page 7 and Fig. 5a,b:

"In the E1RF structure, the conserved positively charged residues Lys356 and Lys359 project their side chains into the E1HD channel but their ϵ -amino groups are at least 4Å from the 5' ssDNA phosphate backbone (Fig. 5a,b)."

Lys359 is not visible in either of these figures. Considering the low resolution of the structure, the authors should also clearly show the density for these side chains so that the reader can judge the reliability of the rotamer assignments and distance claims.

Page 8:

"A range of amino acid substitutions tested at position 352 all showed reduced DNA unwinding activity."

This sentence is missing a reference to figures or previously published data.

Fig. 7b:

It is very difficult to see the described changes in footprinting between WT and E1 mutants in fig. 7. The authors could guide the reader better to the specific differences they think they see or try to show the data in a different way. Based on the data currently presented, this reviewer is not convinced there are significant alterations in the footprinting patterns between WT and E1 mutants.

Fig. 5b,c:

The authors still do not show side chain densities for lysine 310-E and asparagine 352-D despite stating to have amended it in the figure. It is key to show the density clearly (in the zoomed view) so that the reader can judge the quality of the EM map in this area and the confidence in the rotamer assignment of these apparently critical side chains.

The figure naming scheme is not consistent. The authors use both supplementary figure and extended data figure and should settle on one convention.

Movie:

As mentioned last time, a movie showing an overview of the structure and cryoEM map, with a zoomed view of key regions, would be extremely helpful and valuable. The authors state that this task would require professional movie makers. However, standard cryoEM visualization software such as UCSF Chimera and ChimeraX is perfectly equipped to handle this task. The authors may want to look at movies in the following papers: Yan et al. Nature 2019, Goswami et al. Nat. Comms 2018.

Reviewer #3 (Remarks to the Author):

The authors have answered all the raised points in a satisfactory way and the article improved substantially. Data are presented in a clearer manner and I support publication.

Response to the reviewers' comments

We thank the reviewers for the additional comments. Please see below our response. The reviewers' comments are in black and our response in blue.

Reviewer #1 (Remarks to the Author):

The manuscript is much improved. I would have liked a marked up version - without this it was difficult to know what changed. The discussion is more inclusive and more informative. Once the comments below are addressed, I would support publication:

Abstract "The 3' single stranded DNA is pulled through the helicase by the rotational height-adjusted movement of the AAA+ DNA-binding β -hairpins". The section in bold is correct, but not easy to understand for the standard reader. This should be made more accessible.

"The 3' single stranded DNA interacts with the six helically-arranged AAA+ DNA-binding β -hairpins and is pulled through the helicase as the hairpins move in a cycle from top to bottom positions."

Abstract / discussion "while a cyclical movement of collar domain subunits optimises the use of energy from nucleotide hydrolysis for strand separation". I am not sure that this has been demonstrated. Has the energy been calculated to demonstrate that? If it is a speculative model, then this should be indicated.

Yes, the model is speculative, based on our structural observations and modelling of the protein conformational wave around the helicase ring. In the abstract and discussion, we have now made it clear that this is a speculative model.

In the abstract the final sentence now reads: "Pulling of the RF against the collar ring separates the base-pairs, while modelling of the conformational cycle suggest an accompanying movement of the collar ring has an auxiliary role, helping to make efficient use of ATP in duplex unwinding."

The relevant segment in the discussion now reads (lines 388-394): "Using the E1RF structure we modelled an entire conformational cycle of the helicase (Fig. 7, Supplementary Movie 2) and this analysis showed that the tilt and elevation of the collar ring follows a wave-like motion around the subunits, providing an auxiliary "push" against the RFJ. We suggest that base pair separation could be assisted by a once-per-revolution "power-stroke" directly coupled to the ATPase cycle, allowing E1 to make efficient use of the energy of ATP hydrolysis for translocation coupled DNA unwinding."

Introduction "In cells the principal replication helicase". Usually it is called replicative helicase not replication helicase.

The correction has been made.

Figure 7 – Why did the authors use a low concentration of E1. In their recent NAR publication, they observed clear protection of the RFJ only at higher protein concentrations. Overall the protective effect appears rather minor, so maybe this is better placed in the supplementary figures. It is likely that Hydroxy-radical footprint is not ideal to detect these rather weak and transient interactions, or that additional amino acids participate in this process

We agree with the reviewers that the effects on protection are small, but they are measurable. In general, this technique is well suited to probe DNA interactions made by high affinity and sequence-

specific DNA binding proteins, but more technically challenging for a helicase. We believe that the experiments have been carefully performed and analysed, to provide accurate information. We have placed the figure in the supplementary information, in accord with the suggestion of the reviewer. The results section and figure numbering in the manuscript have been altered accordingly. Please see the modifications on page 7 and 9 (results) and an amended sentence in the discussion.

It is worth noting, as discussed in the revised version, that there are parallels with the results from the Yeeles' lab (Baretić, D. *et al. Mol. Cell* 78, 926-940 (2020)), where hypothesised dsDNA binding mutants in Csm3/Tof1 showed no or only small defects in *in vitro* DNA replication.

In these experiments the ratio of protein to DNA (6:1) is critical and not necessarily the absolute concentration. We chose to use a slight excess (10%) of protein (*i.e.* 6.6 E1 molecules per DNA fork) and *confirmed that all DNA was bound using an agarose gel shift assay*. We actually used here a higher concentration of E1 protein (16 μ M) to ensure complete binding.

Reviewer #2 (Remarks to the Author):

In this revised manuscript, the authors address some but not all concerns from the original submission. The following concerns remain or arise from the revision:

Page 4:

“Here we present a near-atomic resolution (3.9 Å) cryo-EM structure of E1 revealing clearly for the first time all arms of the RF and the dsDNA unwinding point for a eukaryotic AAA+ hexameric helicase.”

This sentence remains misleading. Although the E1 structure may have implications for the unwinding mechanism employed by eukaryotic AAA+ helicases, E1 itself is a viral helicase and not a eukaryotic one as this sentence implies.

We agree. The sentence has been modified by the deletion of the word “eukaryotic”. The previous paragraphs have minor modifications to emphasise that both E1 and Mcm2-7 are based on the AAA+ fold.

Page 7 and Fig. 5a,b:

“In the E1RF structure, the conserved positively charged residues Lys356 and Lys359 project their side chains in to the E1HD channel but their ϵ -amino groups are at least 4Å from the 5' ssDNA phosphate backbone (Fig. 5a,b).”

Considering the low resolution of the structure, the authors should also clearly show the density for these side chains so that the reader can judge the reliability of the rotamer assignments and distance claims.

We disagree that the map is of low resolution, since we have good FSC curves that reliably reflect the reconstruction quality. Possibly, according to the current trend in assessment of resolution, this structure could be assessed to be at near-atomic resolution. At the resolution obtained we were able to identify not only large amino acids but also small ones and to trace the entire polypeptide chain for the AAA+ domains, see Supplementary Figure 2b. The overall quality of the rotamer assignments and correspondence to the EM densities is shown in the validation report, provided as an independent assessment by the PDB EM database, as well as by reports of MolProbity (all-atom structure validation) and the Ramachandran plot (see Table 1).

Nonetheless, we have modified Figure 5b and c to show the quality of the fit of the atomic model to the EM map and correspondence of the side chains to the EM densities. Residues Lys356 and Lys359 are labelled now.

Page 8:

“A range of amino acid substitutions tested at position 352 all showed reduced DNA unwinding activity.”

This sentence is missing a reference to figures or previously published data.

The reference followed the next sentence but has now been moved to immediately link with the sentence highlighted by the reviewer.

Fig. 7b:

It is very difficult to see the described changes in footprinting between WT and E1 mutants in fig. 7. The authors could guide the reader better to the specific differences they think they see or try to show the data in a different way. Based on the data currently presented, this reviewer is not convinced there are significant alterations in the footprinting patterns between WT and E1 mutants.

Please see above the response to reviewer 1. The figure has been moved to the supplementary data. We have also extended our explanation of what we observe to better guide readers. Understanding of the figure should be clearer, while sticking with the accepted norm of showing gel images of the footprints and quantification through lane profiling of gel phosphor images. We hope the reviewer appreciates the technically challenging nature of these experiments.

Fig. 5b,c:

The authors still do not show side chain densities for lysine 310-E and asparagine 352-D despite stating to have amended it in the figure. It is key to show the density clearly (in the zoomed view) so that the reader can judge the quality of the EM map in this area and the confidence in the rotamer assignment of these apparently critical side chains.

The figure has been amended, see the answer to the question related to “Page 7 and Fig. 5a,b” above; and please look at the amended Figure 5 and Supplementary Figure 2b.

The figure naming scheme is not consistent. The authors use both supplementary figure and extended data figure and should settle on one convention.

We thank the reviewer for this comment, and have done corrections.

Movie:

As mentioned last time, a movie showing an overview of the structure and cryoEM map, with a zoomed view of key regions, would be extremely helpful and valuable. The authors state that this task would require professional movie makers. However, standard cryoEM visualization software such as UCSF Chimera and ChimeraX is perfectly equipped to handle this task.

We thank the reviewer for this request and have made a version of this movie, that is submitted as a Supplementary Movie File 2.

Reviewer #3 (Remarks to the Author):

The authors have answered all the raised points in a satisfactory way and the article improved substantially. Data are presented in a clearer manner and I support publication.